# Discovering Robotic Interaction Modes
# with Discrete Representation Learning

**Liquan Wang**
The Georgia Institute of Technology
`lwang831@gatech.edu`

**Ankit Goyal**
Nvidia
`angoyal@nvidia.com`

**Haoping Xu**
University of Toronto
`haoping.xu@mail.utoronto.ca`

**Animesh Garg**
The Georgia Institute of Technology
`animesh.garg@gatech.edu`

**Abstract:** Human actions manipulating articulated objects, such as opening and closing a drawer, can be categorized into multiple modalities we define as interaction modes. Traditional robot learning approaches lack discrete representations of these modes, which are crucial for empirical sampling and grounding. In this paper, we present ActAIM2, which learns a discrete representation of robot manipulation interaction modes in a purely unsupervised fashion, without the use of expert labels or simulator-based privileged information. Utilizing novel data collection methods involving simulator rollouts, ActAIM2 consists of an interaction mode selector and a low-level action predictor. The selector generates discrete representations of potential interaction modes with self-supervision, while the predictor outputs corresponding action trajectories. Our method is validated through its success rate in manipulating articulated objects and its robustness in sampling meaningful actions from the discrete representation. Extensive experiments demonstrate ActAIM2's effectiveness in enhancing manipulability and generalizability over baselines and ablation studies. For videos and additional results, see our website: https://actaim2.github.io/.

**Keywords:** robot manipulation, discrete representation learning, interaction mode, self-supervised

## 1  Introduction

Humans exhibit an exceptional aptitude for manipulating articulated objects by utilizing prior knowledge and learning through imitation. Generally, the outcome after manipulating the articulated objects is categorical such as opening or closing the door. Motivated by this cognitive process, our study seeks to develop a discrete representation of object affordance by merging behavior cloning with interaction mode identification. We focus on objects with multiple moving parts to explore a variety of distinct and meaningful outcomes, which we define as *interaction modes*. These interaction modes can be represented as a discrete set of options from which the agent can sample during inference to determine the appropriate interaction mode for the object. Notably, we characterize interaction mode as an affordance property of the object itself which can be learned from observation data. To transfer such discrete interaction modes into robotic action, an action predictor is learned from play data containing these diverse modes. Therefore, we argue that the manipulation policy is a joint distribution over both interaction mode selector and action predictor.

The robotics field is replete with studies addressing such policy decomposed into modes (or skills) and action distribution. However, most behavior cloning approaches [1, 2, 3, 4] require expert data for task representation supervision, especially using language description as the task representation such as RLBench [5], Calvin [6], SayCan [7], etc. This assumes that the distribution of the interaction mode is known and can be represented as language prompting. Other works learn the interaction mode

8th Conference on Robot Learning (CoRL 2024), Munich, Germany.

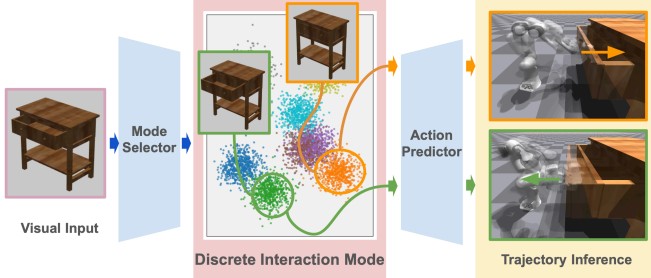

**Figure 1:** ActAIM2 identifies meaningful interaction modes such as open and close drawers from RGB-D images of articulated objects and robots. It represents these modes as discrete clusters of embeddings and trains a policy to generate control actions for each cluster-based interaction.

(skill) distribution using unsupervised reinforcement learning [8, 9, 10, 11] with self-supervised intrinsic reward as weak supervision for task representation. Moreover, there are other interaction mode learning approaches [12, 13, 14] which learn a distinguishable interaction mode (skill) prior. However, the distribution of interaction modes (skills) learned in these works does not specifically map to correspondent outcomes, indicating that the agent cannot sample skills with reasonable and limited options. Therefore, these works fail to find a structural and disentangled space of skill for an agent to sample the interaction mode discretely based on observation. To solve the issues above, we require that the policy 1) captures various interaction modes without necessitating expert labels or privileged information and 2) allows the agent to make discrete choices within this space, reflecting the finite interaction modes available in a given scene.

We introduce ActAIM2, which splits the policy into a discrete mode selector using a Gaussian Mixture Model for discrete sampling and an action predictor that processes sampled task embeddings to predict corresponding action sequences, trained via behavior cloning. Both components utilize self-supervised play data from a dataset collected through adaptive data collection and heuristic grasping, ensuring a balanced representation of interaction modes. Building upon ActAIM's method [12], our data collection does not use privileged information like reward functions or part segmentation and enhances realism by employing a complete robot instead of a floating gripper. To summarize, our contributions are threefold: First, we introduce ActAIM2, a self-supervised learning approach that enables discrete sampling across different interaction modes. Second, We devise a novel data collection methodology and have constructed a dataset with diverse interaction modes for model training. Finally, we thoroughly evaluate our model against a spectrum of generative models and behavior cloning agents, demonstrating ActAIM2's superior performance over existing baselines and the capability of performing the discrete sampling.

## 2   Related Work

***Articulated object manipulation*** – The manipulation of articulated objects is challenging due to their complex geometries and kinematics, often understood through partially observed data like images. Common methods deduce the object's kinematic structure via passive observation [15, 16, 17] or interactive perception [18, 19]. These techniques are crucial for modeling articulated objects in robotics and incorporating these models into planning. While imitation learning relies on expert demonstrations [20, 21], it is limited by the need for extensive and costly data collection. Conversely, recent research explores obtaining actionable visual priors through direct interaction with objects [22, 23, 24], focusing on their geometric and semantic properties. Additionally, learning-based methods utilizing simulation supervised visual learning, and visual affordance learning are emerging [23, 24, 25]. ActAIM2 addresses these challenges by demonstrating the stability of our model in the experiments section.

***Transformer-based policies*** – Many recent works have proposed Transformer-based policies for robotic control. For behavior cloning, Transformers model conditional action distributions [26, 3] and encode observations into latent vectors [2]. They also process language commands to specify and generalize tasks, demonstrating flexibility and robustness [27, 28, 29]. Additionally, Transformers are utilized to learn latent 3D representations from scenes, building on their success in computer vision

[30, 31, 32], and to process voxelized scenes from RGB images and point clouds [2, 33]. However, challenges remain, such as high memory demands for high-resolution voxelizations that slow training, and susceptibility to noisy data when re-rendering views from point clouds [3]. Transformers also map language and visual inputs directly to robot actions, ranging from basic attention layers to complex pre-trained models [27, 34]. ActAIM2 builds on this foundation using the RVT model but innovates by integrating a novel discrete interaction mode representation into the skill language prompting.

*Unsupervised Skill Learning* – Unsupervised skill discovery enables agents to learn distinct behaviors without a reward function but often struggles with inadequate state space exploration when using variational Mutual Information maximization [35, 36]. This limitation hinders its effectiveness for complex tasks. Some strategies counteract this by focusing the learning on smaller, Euclidean spaces to diversify movements [37], although this often restricts the learning to navigation and simple coordinate-based tasks. Alternative approaches propose auxiliary exploration mechanisms and novel training methods to enhance state space exploration [36, 38]. ActAIM2 addresses these issues by introducing the term "interaction mode," defined by significant visual changes from changes in the degrees of freedom (DoF) of an articulated object. By maximizing the mutual information loss and using contrastive evaluation of visual changes, ActAIM2 encourages the model to identify discrete interaction modes and develop a disentangled latent space for effective sampling.

## 3 Problem Formulation

The problem we are solving is how to use a parallel-jaw gripper robot to manipulate various articulated objects and generalize such skills among all types of articulated objects. We adopt a two-phase methodology to enhance a robot's ability to manipulate articulated objects: data collection and model training, inspired by [12]. We employ a structured predefined action primitive, executing a series of actions across four heuristic phases (initiation, reaching, grasping, manipulation) to gather a comprehensive dataset of observations $O_i$, including RGBD images and multi-view camera positions, without relying on predefined inputs. Within each action phrase, we define the action $a$ as the keypose $(\mathbf{p}, \mathbf{R}, \mathbf{q})$ of the parallel-jaw gripper similar to [5]. The key pose is defined as the concatenation of the 3 terms, which are $\mathbf{p} \in \mathbb{R}^3$ as the gripper position, $\mathbf{R} \in SO(3)$ as the gripper rotation quaternion, and $\mathbf{q} \in \{0, 1\}$ as the binary parameter indicating whether the gripper is open or close.

Our model training aims to uncover the policy's distribution $\mathbb{P}(a|o)$, with $o$ representing the observation and $a = (\mathbf{p}, \mathbf{R}, \mathbf{q})$ the action, through a decomposition strategy that reconfigures the action distribution as:

$$\mathbb{P}(a|o) = \int \underbrace{\mathbb{P}(a|o, \epsilon)}_{\text{action predictor}} \underbrace{\mathbb{P}(\epsilon|o)}_{\text{mode selector}} \, d\epsilon \qquad (1)$$

The mode selector $\mathbb{P}(\epsilon|o)$, contrasting with ActAIM [12]'s Gaussian space, utilizes a mixture of Gaussian distributions to define distinct interaction modes, facilitated by a discrete, latent space $\epsilon \in \mathbb{Z}$ for enhanced action prediction and mode selection.

## 4 ActAIM2: Robotic Interaction Mode Discovery

We aim to derive the policy $\mathbb{P}(a|o)$ from Equation 1, employing the robot as the agent within an environment populated by various articulated objects. Motivated by [12], our preliminary step was to gather offline, self-supervised data via simulation, which served as the foundation for training the policy $\mathbb{P}(a|o)$. Guided by Equation 1, we dissected the target policy into the action predictor $\mathbb{P}(a|o, \epsilon)$ and the mode selector $\mathbb{P}(\epsilon|o)$. For training efficacy, the action predictor $\mathbb{P}(a|o, \epsilon)$ and mode selector $\mathbb{P}(\epsilon|o)$ were pre-trained individually before collectively fine-tuning the overarching pipeline $\mathbb{P}(a|o)$.

## 4.1 Iterative Data Collection

We collect trajectory data $T_j = \{(a_i, O_i) | i = 0, 1, 2, 3\}_j$ where $O_i$ are RGBD observations from a configuration of five cameras encircling the articulated object, and $a_i = (\mathbf{p}, \mathbf{R}, \mathbf{q})_i$ represents the key pose and state of the gripper. Inspired by ActAIM [12], we use a similar GMM adaptive method to collect diverse interaction modes. To collect the data with self-supervision, for each trajectory, characterized by the initial observation $O_j^{init} = O_{0j}$ and final observation $O_j^{final} = O_{3j}$, we utilize a pre-trained image encoder $\mathcal{E}_O$ to transform the image observations into a latent vector $v$. The task embedding $z_j$ for each trajectory $T_j$ is defined as follows:

$$z_j = v_j^{init} - v_j^{final} = \mathcal{E}_O(O_j^{init}) - \mathcal{E}_O(O_j^{final}) \tag{2}$$

To determine the success or failure of a manipulation, we introduce a threshold $\bar{z}$, defining a trajectory $T_j$ as successful if $z_j > \bar{z}$. Details of the dataset are presented in Appendix 7.4.

## 4.2 Learning Interaction Modes with Discrete Representations $\mathbb{P}(\epsilon|o)$

***Learning Generative Model using GMM Prior*** –Based on Equation 1, we aim to identify an effective mode selector to generate a relevant task latent space $\epsilon \in \mathbb{Z}$. Inspired by the concept of visual affordances in [39], we learn such latent space encapsulates all conceivable future states using the conditional Gaussian Mixture Variational Autoencoder (GMVAE) generative model.

To learn the mode selector as prior $\mathbb{P}(\epsilon|o)$, we use the data from our collected trajectory and select data $(O^{init}, O^{final})_j$ for mode selector training. Knowing that $z$ from Equation 2 contains the complete information of final states given initial observation, We pre-compute the ground-truth label $z$ as our learning target and regard the initial observation $O^{init}$ as the conditional variable to generate the task embeddings. We aim to learn a generative model which predicts all possible task embeddings $z$ from the conditional initial observation $O^{init}$. Taking inspiration from [40], we construct the generative model as a Transformer-based Conditional GMVAE, as shown in Figure 2a. We defined our inference process as $O^{init} \to (c, y) \to x \to \epsilon$ where $c$ is a categorical variable with $p(c) = \text{Multi}(\pi)$ and $y$ is a Gaussian distribution with $p(y) = \mathcal{N}(0, \mathbf{I})$ which jointly forms a Gaussian Mixture distribution as the prior. We expect the output learned $\epsilon$ would have a similar distribution as the ground-truth task embedding $z$ since $z$ can be represented as a Mixture of Gaussian distribution fixing the initial observation (fixing the initial object and object state). Formally, we write the distribution of the variable under the following process; for simplification, we denote $O^{init}$ as $O^i$,

$$p_{\xi,\beta}(\epsilon, x, y, c|O^i) = p(y)p(c)p_\xi(x|y, c, O^i)p_\beta(\epsilon|x, O^i) \tag{3}$$

$$p_\xi(x|y, c, O^i) = \prod_{k=1}^{K} \mathcal{N}(\mu_{c_k}(y, O^i), \Sigma_{c_k}(y, O^i)) \tag{4}$$

$$p_\beta(\epsilon|x, O^i) = \mathcal{N}(\mu_\beta(x, O^i), \Sigma_\beta(x, O^i)) \tag{5}$$

where $K$ represents the number of mixture components, a hyper-parameter within the training regime, and $\mu_{c_k}, \Sigma_{c_k}, \mu_\beta, \Sigma_\beta$ denote networks to be trained where $\xi$ and $\beta$ are the parameters of these networks. In our implementation, $\mu_{c_k}, \Sigma_{c_k}$ are instantiated as multi-layer ResNet [41] architectures, and $\mu_\beta, \Sigma_\beta$ as a transformer with 4 self-attention layers to enhance the stability of reconstruction. To improve the training stability, we model the categorical distribution $p(c)$ by the Gumbel-Softmax distribution [42].

***Mode Selector Training Loss*** –The conditional-GMVAE generative model is optimized using the variational inference objective, combining reconstruction loss with the log-evidence lower bound (ELBO) loss. The ELBO loss is expressed as:

$$\mathcal{L}_{ELBO} = \mathbb{E}_q \left[ \frac{p_{\xi,\beta}(\epsilon, x, y, c|O^i)}{q(x, y, c|\epsilon, O^i)} \right] \tag{6}$$

where the proxy posterior $q(x, y, c|\epsilon, O^i)$ is approximated as $q(x, y, c|\epsilon, O^i) = \prod_i q_{\psi_x}(x|\epsilon, O^i)q_{\psi_y}(y|\epsilon, O^i)q_{\psi_c}(c|x, y, O^i)$, with $\psi_x, \psi_y$ representing the parameters of the

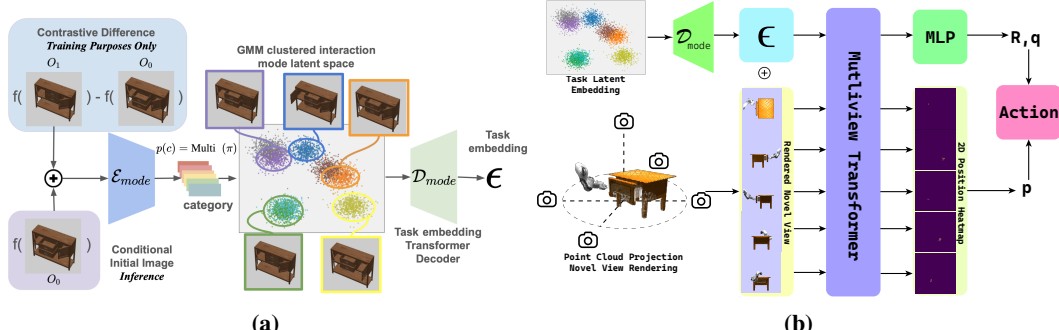

**(a)**
**(b)**

Figure 2: **(a) GMM Model Selector** The mode selector, a generative model, processes the differences between the initial and final image visual embeddings as generated data, using the initial image embeddings as the conditional variable. **(b) Behavior Cloning Action Predictor** Interaction mode $\epsilon$ is sampled from latent space embedding from model selector. 5 Multiview RGBD observations from circled cameras are back-projected and fused into a color point cloud to render novel views. Rendered image tokens and interaction mode token are contacted and fed through a multiview transformer to predict action $a = (\mathbf{p}, \mathbf{R}, \mathbf{q})$.

$q_{\psi_x}$ and $q_{\psi_c}$ networks. Based on the decomposition, We refer to the terms in the lower bound as the reconstruction term, conditional prior term, $y$-prior term and $x$-prior term respectively. The prior term for variables $c, y, x$ is computed as the Kullback-Leibler (KL) divergence, which penalizes the difference between the learned latent variable distribution and the prior distribution. We formalize the reconstruction term as the L2 loss between task embedding predictions $\epsilon$ and ground-truth $z$, written as $\mathcal{L}_{reconstruct} = ||\epsilon - z||^2$ in practice. Mathematical details of the conditional GMVAE are presented in Appendix 7.5.1.

## 4.3 Supervised Action Predictor Learning $\mathbb{P}(a|o, \epsilon)$

Our final objective is to infer a sequence of low-level actions $a = (\mathbf{p}, \mathbf{R}, \mathbf{q})$ from the current observation $O$ given the predicted task representation $\epsilon$, ensuring the action sequence effectively accomplishes the articulated object manipulation task while aligning the specific given interaction mode $\epsilon$. As shown in Figure 2b, The model inputs RGB-D images from encompass multi-view cameras, the present state of the robot gripper, and the task latent embedding $\epsilon$. From the data collection phase, we collect successful manipulation trajectories $T_j$. We decomposed $T_j$ to individual key-frame actions and observations $(a_i, O_i)_j \in T_j \in D$ as data points for training. To this end, we propose a multi-view transformer architecture as the basis for our behaviour cloning agent, aimed at learning the action distribution $\mathbb{P}(a|o, \epsilon)$.

*Novel View Rendering and Multiview Transformer* – Building upon the RVT [3] approach, we utilize novel view rendering from RGB-D multi-view cameras as our visual observation, strategically positioning five cameras around the robot and articulated objects to generate a merged RGB point cloud. This cloud is normalized to the scene center and projected onto orthogonal image planes, creating novel views from the top, front, behind, left, and right, each incorporating RGB color, XYZ position, and depth channels as model inputs. These rendered views are processed by a multiview transformer model, where images are patchified and encoded with MLPs and positional encoding, similar to the ViT process. Additionally, the task embedding $\epsilon$ and the current state of the gripper are encoded and their features are concatenated with image token representations. The transformer then processes the merged tokens, outputting per-view 2D heatmaps and global features which predict the current state action $a = (\mathbf{p}, \mathbf{R}, \mathbf{q})$.

*Action Predictor Training Loss* – We optimize our action predictor model utilizing a behaviour cloning loss framework. For the position heatmap, cross-entropy loss is employed, with the ground-truth image being synthesized from the 3D ground truth point $\hat{\mathbf{p}}$ through projection onto a 2D orthogonal view, following a Gaussian distribution for spatial smoothing. Similarly, the rotation heatmap is refined using cross-entropy loss for each Euler angle axis, translating the ground-truth rotation $\hat{\mathbf{R}}$ into an analogous one-hot vector representation. The binary classification loss, essentially

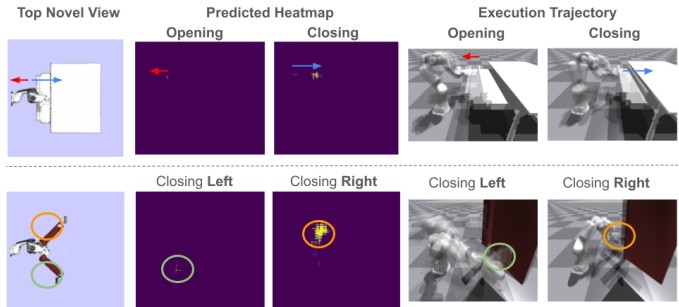

**Figure 3:** Given different task embedding, we see how action predictor produces actions representing distinct interaction modes. Here, we visualize the camera view and the prediction heatmap from the top for object instances. The first row shows heatmaps for pushing and pulling the handle, while the second row shows heatmaps for closing the left or right door. More qualitative results please see the appendix 7.6

a cross-entropy loss, updates the gripper jaw's open-close state $\mathbf{q}$. Accordingly, the comprehensive training loss for the action predictor is articulated as:

$$\mathcal{L}_{action} = \mathcal{L}_{\mathbf{p}} + \mathcal{L}_{\mathbf{R}} + \mathcal{L}_{\mathbf{q}} = CE(\mathbf{p}, \hat{\mathbf{p}}) + CE(\mathbf{R}, \hat{\mathbf{R}}) + CE(\mathbf{q}, \hat{\mathbf{q}}) \tag{7}$$

### 4.4 Training Procedure

In our training process, we jointly train the policy $\mathbb{P}(a|o)$ as formulation:

$$\mathbb{P}(a|o) = \mathbb{P}(a|o, \text{sg}[\epsilon]) \, \mathbb{P}(\epsilon|o) \tag{8}$$

In this context, $\text{sg}[\cdot]$ stands for the stop-gradient operator, which stops the flow of partial derivatives to the next network layers. During the training phase, we include the combined total loss from both the action predictor (behavior cloning loss) and the mode selector (ELBO loss), represented as $\mathcal{L}_{total} = \mathcal{L}_{action} + \mathcal{L}_{ELBO}$.

## 5 Experiments

Our experimental setup is designed to assess how well the proposed method, ActAIM2, performs in several important areas: 1) ActAIM2 proficiently handles various interaction modes, adapting to differences across object instances and categories. 2) Utilizing Gaussian Mixture Model (GMM) based priors facilitates a more structured latent space, enabling targeted searches for specific samples within the interaction modes. To evaluate ActAIM2, we report the average success rate, sample success rate, and the average reward achieved during interaction mode grounding iterations.

### 5.1 Experimental setup

Leveraging the success of previous studies [43, 2, 12], we use articulated objects from the SAPIEN dataset [44] for our experiments. Our training dataset includes six categories: faucets, tables, storage furniture, doors, refrigerators, and switches, while the testing dataset features three new categories: windows, boxes, and safes, with each category comprising 8 unique objects. We employ IsaacGym as our simulation platform [45], where we have designed a custom environment featuring a Franka Emika robot adjacent to an articulated object with 5 cameras circling the object. In our evaluations, we define a successful trajectory as follows: the model takes an initial observation and iteratively predicts a series of actions, executed across four steps. After these steps, we evaluate the object's state. If the Degrees of Freedom (DoF) of any part of the object has changed by more than 30%, we deem the trajectory successful. This approach allows us to assess the efficacy of the robot's manipulation capabilities in dynamically altering the object's state based on the model's iterative predictions.

***Baselines And Ablation Study*** –We compare our results with the following baselines and ablation study. **Data Collection**: Adaptive data collection using a GMM-based heuristic grasping method to sample action sequences. **Where2Act** [43]: Calculates priors for discretized action primitives, using the complete object point cloud instead of segmented movable points. **ActAIM** [12]: Combines a Conditional Variational Autoencoder (CVAE) [46] with a transformer-based action predictor [47] to sample and forecast manipulation trajectories, employing an unrealistically simplified floating gripper.

Table 1: **Robotic Interaction Mode Discovery:** We evaluate our design decisions through baseline comparisons and ablation studies using the sample-success rate (SSR) metric. We find that that ActAIM2 consistently surpasses competing models across various object types.

| Test Set | Seen Objects | | | | | | AVG | Unseen instances | | | | | | AVG | Unseen Cats | | | AVG |
|---|---|---|---|---|---|---|---|---|---|---|---|---|---|---|---|---|---|---|
| SSR % ↑ | | | | | | | | | | | | | | | | | | |
| Data Collection | 12.8 | 8.9 | 9.4 | 16.9 | 10.4 | 8.9 | 11.2 | 11.4 | 9.5 | 14.0 | 13.2 | 11.9 | 12.9 | 12.2 | 20.4 | 17.3 | 15.0 | 17.6 |
| Where2Act [43] | 33.3 | 7.0 | 7.0 | 17.9 | 12.1 | 4.1 | 13.6 | 33.0 | 13.8 | 19.2 | 16.9 | 13.9 | 15.4 | 18.7 | 15.0 | 16.8 | 15.2 | 15.7 |
| ActAIM [12] | 49.3 | 41.4 | 36.2 | 28.6 | 24.5 | 19.7 | 33.3 | 22.0 | 38.1 | 35.5 | 21.0 | 18.2 | 16.2 | 25.2 | **38.4** | 24.1 | 31.8 | 31.5 |
| Goal-RVT | 58.4 | **48.9** | 51.2 | **72.1** | 23.2 | 33.3 | 47.8 | 40.2 | **43.4** | 39.2 | **65.1** | 18.3 | **25.3** | 38.6 | 32.3 | 23.1 | 33.9 | 29.8 |
| **ActAIM2** | **65.3** | 43.2 | **52.1** | 69.2 | **25.3** | **36.2** | **48.6** | **44.9** | 41.2 | **41.5** | 60.2 | **20.1** | 24.4 | **38.7** | 34.3 | **28.9** | **34.1** | **32.4** |

Table 2: **Mode Sampling Evaluation** ActAIM2 executes actions sampled uniformly across 8 clusters, while Goal-ActAIM and Goal-RVT use manually selected goal images as expert labels for mode-specific sampling. The best performance is highlighted in bold, underscoring ActAIM2's consistent improvement across evaluations.

| Test Set | | Seen Objects | | | | | | AVG | Unseen instances | | | | | | AVG | Unseen Cates | | | AVG |
|---|---|---|---|---|---|---|---|---|---|---|---|---|---|---|---|---|---|---|---|
| Algorithm | Mode SSR % ↑ | | | | | | | | | | | | | | | | | | |
| Goal-ActAIM | common mode | 25.3 | 37.5 | 19.3 | 62.9 | 24.3 | 61.2 | 38.4 | 20.3 | 36.3 | 18.2 | 43.0 | 21.3 | 31.4 | 28.4 | 29.3 | 15.2 | 43.5 | 29.3 |
| | rare mode | 24.1 | 16.2 | 11.3 | 28.4 | 7.8 | 17.6 | 17.6 | 12.4 | 14.5 | 9.5 | 10.5 | 5.0 | 13.1 | 10.8 | 16.0 | 7.8 | 37.5 | 20.4 |
| Goal-RVT | 1st mode goal | 42.4 | **57.4** | 76.5 | 75.4 | 44.9 | **65.4** | 60.3 | 40.2 | **48.8** | **72.4** | 74.3 | 30.2 | 50.3 | 52.7 | 35.5 | **35.5** | 49.2 | 40.1 |
| | 2nd mode goal | 17.2 | 28.3 | 31.2 | 70.4 | 20.1 | 34.2 | 33.6 | 24.3 | 20.4 | 15.2 | 64.2 | 10.1 | 20.3 | 25.8 | 10.4 | 5.1 | 5.1 | 6.9 |
| VOVAE-RVT | 1st mode vector | 64.4 | 44.9 | 56.3 | 64.3 | 27.4 | 35.2 | 48.75 | 45.6 | 39.3 | 40.2 | 51.8 | 29.1 | 30.2 | 39.4 | 34.2 | 26.9 | 36.4 | 32.5 |
| | 2nd mode vector | 0.0 | 0.0 | 0.0 | 0.0 | 0.0 | 0.0 | 0.0 | 0.0 | 0.0 | 0.0 | 0.0 | 0.0 | 0.0 | 0.0 | 0.0 | 0.0 | 0.0 | 0.0 |
| **ActAIM2** | 1st mode cluster | **95.4** | 45.3 | **83.4** | **80.4** | **58.8** | 65.4 | **71.5** | **70.4** | 45.4 | 72.4 | **78.5** | **40.3** | **58.9** | **61.4** | **43.8** | 34.9 | **65.8** | **48.2** |
| | 2nd mode cluster | 15.4 | 24.5 | 20.4 | 60.4 | 10.2 | 31.2 | 27.0 | 10.1 | 15.8 | 15.2 | 58.3 | 5.0 | 20.3 | 20.9 | 10.4 | 5.1 | 5.1 | 6.9 |

**Goal-RVT**: A supervised version of ActAIM2 that uses directly provided goal images, bypassing GMM prior sampling, to feed into the action predictor, demonstrating comparable performance to other supervised methods. **VQVAE-RVT**: Inspired by Genie [48], replacing the interaction mode selector with a VQVAE codebook, using a codebook size equal to the GMM cluster number in ActAIM2, and sampling task embeddings discretely during tests.

*Evaluation Metrics* –We conduct two types of evaluations to assess the ability of models for successful interaction and discrete sampling, testing across three object categories: Seen Objects (from the training set), Unseen Instances (same categories as Seen but different instances), and Unseen Categories (from outside the training set). **Interaction Mode Discovery** metric assesses the average sample success rate (SSR) defined as the ratio of successful trajectory samples to total samples. For Goal-RVT, expert-labeled goal images are used to ensure viable trajectories, whereas ActAIM2 samples from all 8 clusters to calculate SSR. Moreover, we measure SSR for dominant interaction modes within the selected 2 clusters as **Mode Sampling Evaluation**. This indicates that the evaluated method is required to have at least 2 interaction modes discovered and distributed in separate sample options. For ActAIM2 and VQVAE-RVT, we sample and execute actions from 8 distinct clusters, documenting the success rate of the primary interaction mode in the first and second clusters. Goal-RVT and Goal-ActAIM follow a similar approach but use ground truth goal images instead of sampled clusters, adjusting for changes in DoF.

## 5.2 Discussion of Results

*Interaction Mode Discovery* – Table 1 shows that ActAIM2 often outperforms baselines in interaction success rates for both trained and unseen instances. Where2Act and ActAIM use a simplified scenario with a heuristic floating gripper, whereas ActAIM2 utilizes a complete Franka Emika robot setup but still performs a higher success rate. This means ActAIM2 performs well even when it has to discard trajectories due to unreachable states or collisions. This highlights its ability to accurately predict gripper positions and efficiently sample from a learned latent space $\mathbb{Z}$, surpassing the performance of Goal-RVT's broader goal image approach. We did not report the VQVAE-RVT results here since VQVAE-RVT outperforms around 5% in each test compared to ActAIM2 averagely. However, despite the average sample success rate, we show that VQVAE-RVT does not meet our requirement for discrete sampling in the following two experiments.

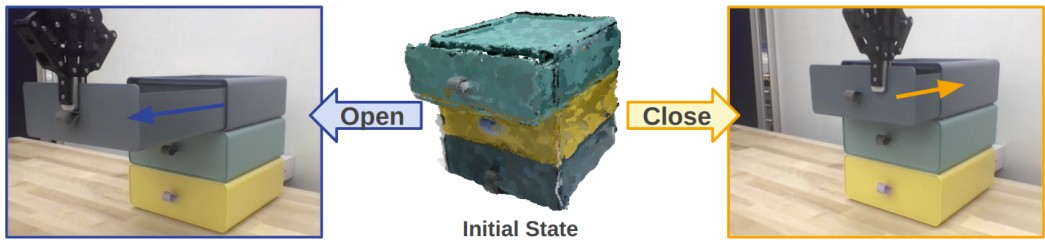

**Figure 4:** The figure illustrates the drawer manipulation task conducted by the Kinova Gen2 robot arm. The task involves interacting with a three-drawer shelf, starting from an initial half-open state (center). The robot executes two modes: opening (left) and closing (right) the drawers, with arrows showing the gripper's movement direction during each interaction.

*Mode Sampling Evaluation* – ActAIM2 and Goal-RVT were evaluated for their ability to distinguish between interaction modes. ActAIM2 proved stable and successfully identified the primary interaction mode across various categories and conditions, matching Goal-RVT's performance even for a secondary mode. Larger objects like doors and refrigerators consistently showed successful interaction trajectories, demonstrating ActAIM2's adaptability to different object sizes and complexities. In contrast, VQVAE-RVT struggled to offer a viable second dominant interaction mode. Our tests showed that VQVAE-RVT tends to produce similar action heatmaps across different code vectors, complicating the identification of specific interaction modes. These issues are further explored in our reinforcement learning experiments on interaction mode grounding.

### 5.3 Real Robot Experiments

We evaluate ActAIM2 in a real-world setup using visual data. A Kinova Gen2 robot arm interacts with a small shelf featuring three drawers, performing opening and closing actions on half-open drawers (see Figure 4). An Azure Kinect RGB-D camera captures the scene, and after calibrating the camera extrinsic, we process point clouds with ActAIM2. The robot's gripper grasps the drawer edge to pull (open) or push (close) the drawers. We set three initial states with the top, middle, or bottom drawer half-open, introducing task variation by moving the shelf's position and randomly selecting grasp points. ActAIM2 is pre-trained on simulated data and fine-tuned with 60 real-world trajectories collected via expert control using four action primitives. After fine-tuning, ActAIM2 achieved a 75% success rate for both pushing and pulling tasks, accurately classifying interaction modes. Sampling from the 8 clusters representing interaction modes, specific clusters had success rates over 87.5%, mirroring the mode sampling evaluation and demonstrating the benefit of incorporating human demonstration data. Some failures occurred due to the gripper missing the drawer edge when the object's position shifted slightly, suggesting the need for more detailed tuning and increased variation in training to improve object positioning accuracy. For more detailed information on the real robot experiments, please refer to Section 7.1 and our website.

## 6 Conclusion

ActAIM2 marks a major advancement in self-supervised learning for robotic control, enabling robust discrete sampling across diverse interaction modes. Our specialized data collection and dataset lay a solid foundation for future enhancements. Extensive comparisons show ActAIM2's superior performance over existing models, enhancing discrete interaction mode learning and strengthening self-supervised discovery techniques for practical applications.

**Limitations.** ActAIM2 offers a promising self-supervised method for learning interaction modes, but balancing success rates across modes remains challenging, especially for less common interactions. Iterative data collection targeting these modes could improve performance. Additionally, ActAIM2 assumes interactions are predefined simple action sequences, which limits its application to complex multi-stage manipulations. For a detailed discussion on assumptions and future work, see Appendix 7.3.

**Acknowledgments**

We would like to express our sincere gratitude to Professors Humphrey Shi and Sonia Chernova from the Georgia Institute of Technology for providing the resources that made this research possible. Their generous support, through access to computation resources and necessary hardware, was invaluable to the success of this project. We also extend our thanks to the Vector Institute for providing computation resources. Additionally, we thank the members of PairLab for their helpful discussions and feedback, which contributed to shaping the direction of this research.

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

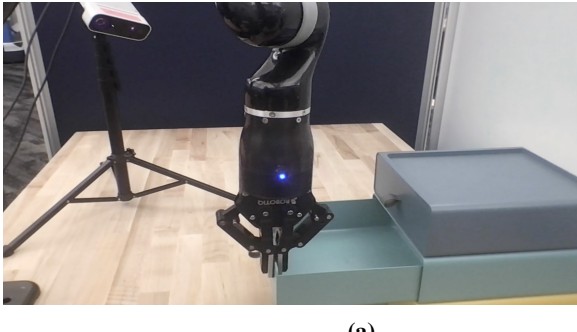 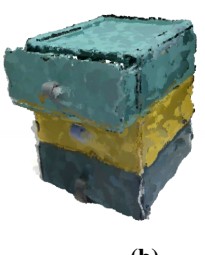

(a)                                                                          (b)

**Figure 5: (a) Real World Setup:** In the image, we demonstrate the real-world experiment setup using a single RGB-D camera (Azure Kinect) to capture visual information and a Kinova robot equipped with a parallel jaw for interacting with an articulated shelf object.
**(b) Shelf Object Point Cloud Illustration:** We present the point cloud of the shelf object with three movable drawers, extracted from the RGB-D camera.

## 7 Supplementary

### 7.1 Real Robot experiments

In this section, we evaluate ActAIM2 using real visual sensory data by training and testing the model in a real-world setup. For further insights into the model's performance, we have included attached videos, which are also available on our website.

***Real World Setup*** – We employ a Kinova Gen2 robot arm with a spherical wrist, 7 degrees of freedom, and a parallel jaw gripper (Robot Type: j2s7s300), placed on a table alongside the articulated object shown in Figure 5a. The scene is captured using a statically mounted Azure Kinect (RGB-D) camera in a third-person view. We first calibrate the robot-camera extrinsics and transform the perceived color point clouds to the robot's base frame before processing them with ActAIM2. During evaluation, based on the target pose predicted by ActAIM2, we utilize the MoveIt motion planner to direct the robot to the target pose, incorporating trajectory generation and feedback control.

***Tasks*** – We focus on manipulating a small shelf with three drawers showed in Figure 5b, defining two interaction modes for each half-open drawer: opening and closing. For the opening interaction, the gripper manually grasps the edge of the drawer and pulls it out. Conversely, for closing, the gripper grasps the edge and pushes the drawer in. We utilize a small shelf with three drawers as the articulated object and set three plausible initial states: the top, middle, and bottom drawers are each set to half-open. Task variation is introduced by altering the position of the small shelf on the table.

***Data Collection and Training*** – Our evaluation focuses on a single task: opening and closing a half-open drawer on a small shelf. Initially, ActAIM2 is pre-trained using domain randomization on our simulated dataset. We currently employ ActAIM2 trained on the single-view dataset referenced in Table 4. For expert demonstration, we collect real-world interaction data via expert control to either open or close the drawer. The robot control strictly adheres to the simulator setup, utilizing four predefined action primitives: initializing, reaching, grasping, and interacting. To introduce variation in the training trajectories, each expert-controlled interaction randomly selects different parts of the drawer for the robot to grasp. Throughout these interactions, the articulated object remains stationary. Simultaneously, RGB-D images are captured as the robot moves to the target pose, creating a dataset of RGB-D frames paired with target pose annotations. For each initial state (top, middle, and bottom drawers half-open), we collect 10 expert data sets for each mode, totaling 60 trajectories (10 pushing and 10 pulling trajectories for each state).

***Results*** – We initially trained ActAIM2 on simulated data and later fine-tuned it using real-world trajectories. This process yielded a notable success rate of 75% for both the pushing and pulling tasks involved in drawer interactions. Importantly, ActAIM2 accurately classifies the interaction modes of pushing and pulling, demonstrating a well-aligned discrete representation. To assess the performance,

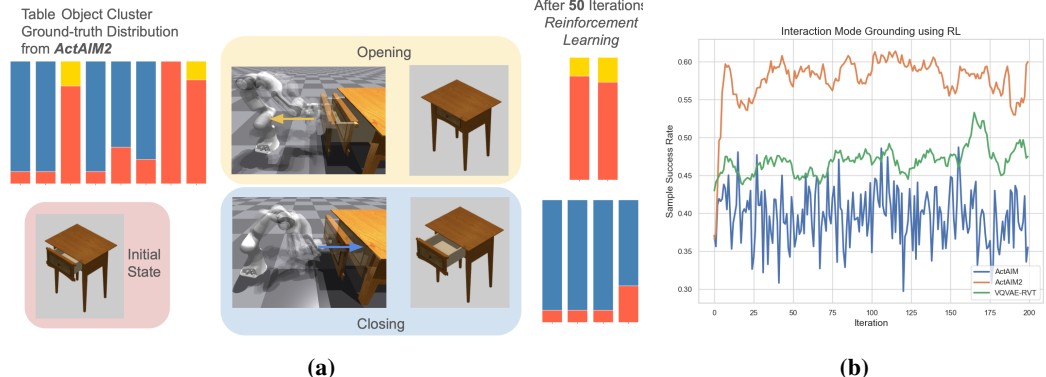

(a)                                 (b)

**Figure 6: (a) Visualization of Reinforcement Learning on ActAIM2:** ActAIM2's grounding through reinforcement learning is demonstrated, highlighting successful openings (yellow), closings (blue), and failures (red) across eight interaction clusters. After 50 iterations, clusters are accurately identified, enabling consistent trajectory generation.
**(b) Reward Optimizing Plot:** This plot displays reward optimization over iterations (x-axis) and average reward (y-axis, indicative of success rate for the targeted mode). ActAIM2's structured latent space shows clear advantages in sampling and convergence efficiency compared to ActAIM [12] and VQVAE-RVT.

we sampled from each of the 8 clusters designed to represent distinct interaction modes—either opening or closing the drawer. Clusters specific to pushing and pulling were found to have success rates exceeding 87.5%, mirroring the mode sampling evaluation. This improvement is largely due to incorporating human demonstration data, which helps correct any imbalance in interaction mode representation during training.

However, in our real robot experiments, we observed some failure cases where the gripper failed to grasp the edge of the drawer. These failures are linked to task variations, specifically the slight movement of the object along the x-axis, which led to the gripper failing due to the agent's incorrect determination of the object's position. To address this issue, we suggest implementing a more detailed coarse-to-fine tuning approach on the vision data and training with increased variation to improve accuracy in object positioning.

## 7.2 Interaction Mode Grounding with ActAIM2 using Reinforcement Learning

We further explore ActAIM2's capability as a prior in reinforcement learning to achieve specified interaction modes given a goal, iterating through all plausible interaction modes to effectively ground and distinguish each mode. In our setup, we used an unseen table (ID:20411) as the articulated object with a sparse reward function defined as $r = 1$ if $|d_i - d_g| < d_\epsilon$ else 0, where $d_i$ is the selected $i$th degree of freedom (DoF), $d_g$ is the target DoF, and $d_\epsilon$ is the threshold. Reinforcement learning was employed using ActAIM2 as the prior, framing the task as a Multi-Arm Bandit problem [49], which narrows the sampling space and enhances learning efficiency. For comparison, we used ActAIM and RVT-VQVAE as the prior in the baseline experiments, applying the same reward function but updating the sample task embedding via the cross-entropy method [50].

Qualitative and quantitative results are visualized in Figure 6a and Figure 6b. Figure 6a shows agents refining their understanding of interaction modes through repeated sampling, leading to self-supervised re-clustering. Figure 6b illustrates that using ActAIM2, the reward—equivalent to the average sample success rate—smoothly increases, whereas ActAIM struggles to ground the task embedding to a specific interaction mode representation, and VQVAE-RVT updates slowly due to its inability to provide distinct interaction mode representations.

## 7.3 Assumptions and Future Research

### 7.3.1 Assumptions

During the data collection phase, ActAIM2 operates under the assumption that: 1) the manipulations are straightforward enough to be captured using a limited set of action primitives such as grasping, pushing, or pulling; 2) an interaction mode is identified upon observing significant visual changes; 3) the interaction modes can be categorized into a few distinct types. A more detailed discussion of these assumptions is provided below.

***Simple Action Space*** – We employ a scripted, self-supervised method to collect actions that encompass diverse interaction modes. The action space is sufficiently simple, focusing primarily on heuristic grasping and random actions. For more complex tasks, such as hammering, washing dishes, or cooking, our current method fails to collect adequate data. Addressing these more intricate tasks would require a more comprehensive and extensive dataset.

***Significant Visual Change*** – Our data collection is entirely self-supervised, devoid of any expert data or privileged information. We define an interaction as successful if it results in a significant visual alteration to the targeted objects. This approach is effective for articulated objects in our studies, such as doors, windows, or tables, which typically remain stationary except for their movable components. However, challenges arise with objects like tools (e.g., hammers, cups, knives), where it is difficult to discern visual changes either in the tools themselves or the targeted objects (e.g., nails, cup holders, or deformable objects). Especially in tasks requiring repetitive actions, like continuously striking a nail or repeatedly wiping dishes, a more nuanced and generalized method is necessary to determine if meaningful interactions are occurring.

***Discrete Interaction Modes*** – Articulated objects, by design, often have limited manipulation options. However, when dealing with other objects such as tools, the number of potential interaction modes significantly increases. The functionality of these objects can be diverse; for example, a hammer might be used not only for hammering but also for hooking or reaching. Even the act of grasping these objects presents countless variations, complicating the task of clustering them into discrete modes.

### 7.3.2 Future Research

Based on the assumptions discussed earlier, we have identified two primary avenues for extending our current research: long-horizon planning tasks and enhancing tool manipulation strategies.

***Long-horizon Planning Tasks*** – Leveraging the discrete representation of interaction modes provided by ActAIM2, we propose its application to long-horizon planning tasks. Examples of such tasks include sequentially opening a table drawer, locating and opening a box within the drawer, and finally pressing a button inside the box. These tasks illustrate the potential of ActAIM2 to serve as a foundational prior, streamlining the process to discrete searches within complex sequences. To ensure the robustness of our approach, it is crucial that the model accurately predicts all feasible interaction modes based on the given scenario.

***Extension to Tool Manipulation Tasks*** – Another direction for expansion involves applying our work to tool manipulation. Here, defining the interaction modes for various tools will be pivotal. A robust dataset specifically tailored for tool manipulation is essential to support this endeavour. Additionally, a more sophisticated scene descriptor is required to effectively determine which objects to manipulate and which to designate as targets. This development would facilitate more nuanced and effective tool interactions in automated systems.

## 7.4  Dataset Generation

### 7.4.1  Iterative Data Collection Method

When collecting data, we employ a strategy of random sampling, subsequently filtering successful actions as determined by our vision model without resorting to any privileged information. Drawing inspiration from [43, 51], we delineate the task of manipulating articulated objects into four fundamental poses: initiation, reaching, grasping, and manipulating. Throughout these stages, we capture the robot's key action poses $a_i = (\mathbf{p}, \mathbf{R}, \mathbf{q})_i$ and RGBD observations $O_i$ from a configuration of five cameras encircling the articulated object. Upon collecting the trajectory $T_j = \{(a_i, O_i)|i = 0, 1, 2, 3\}_j$, we also archive the initial and final observations, $O_j^{init}$ and $O_j^{final}$, respectively, captured from the multi-view cameras with the robot occluded, to facilitate manipulation success evaluation.

We introduced our method of identifying successful interacted trajectories, which can be purely from vision data, specifically the initial and final observation. For each trajectory, characterized by the initial observation $O_j^{init}$ and final observation $O_j^{final}$, we utilize a pre-trained image encoder $\mathcal{E}_O$ to transform the image observations into a latent vector $v$. The task embedding $z_j$ for each trajectory $T_j$ is defined as follows:

$$z_j = v_j^{init} - v_j^{final} = \mathcal{E}_O(O_j^{init}) - \mathcal{E}_O(O_j^{final}) \tag{9}$$

In our implementation, we employ a pre-trained VGG-19 network [52], without the final fully connected layers, to serve as our image encoder $\mathcal{E}_O$. To determine the success or failure of a manipulation, we introduce a threshold $\bar{z}$, defining a trajectory $T_j$ as successful if $z_j > \bar{z}$. It is important to note that this process does not rely on any privileged information. To illustrate the validity of our method, we define the trajectory's success as a $30\%$ change in the ground-truth DoF value. The efficacy of this criterion is validated against the ground-truth DoF values, demonstrating a $97.4\%$ accuracy rate across our training and testing dataset. The collected trajectories must exhibit the diversity of interaction modes of the articulated objects. Thus, we employ three distinct methods of action sampling, as outlined below. The final dataset is a composite of these three methods.

*1. Random Sampling* – We generate play data for manipulation without prior interaction. First, we select an interaction point $p_1 \in \mathbb{R}^3$ on the articulated object, ensuring it lies within the robot's workspace. Subsequently, we sample a uniformly random manipulation rotation $\mathbf{R}_0 \in SO(3)$ and a manipulation position $p_2$ within the valid area, applying filters to exclude any configurations that would result in a collision. The robot's initial position $p_0$ is also determined through random sampling, which is a specified distance from the interaction point $p_1$, ensuring a feasible starting position for the manipulation task. Based on the previous sampling, we define the randomly sampled action sequence as $\{(p_0, \mathbf{R}_0, 0), (p_1, \mathbf{R}_0, 0), (p_1, \mathbf{R}_0, 1), (p_2, \mathbf{R}_0, 1)\}$.

*2. Heuristic Grasping Sampling* – Heuristic grasping sampling is employed to select interaction points on the articulated object to enhance the precision of grasping actions. Utilizing the RGB-D observations, we crop the articulated object and transform it into an RGB point cloud, which undergoes preprocessing with DBSCAN clustering [53], aimed at identifying segments with significant geometric features, such as handles or buttons. After clustering, each segment is analyzed by a pre-trained GraspNet model [54] to generate a set of potential grasps. From this set, grasps with the highest scores are selected, with the grasp point designated as the interaction point and the grasp orientation as the gripper rotation for the trajectory. The initial and manipulation poses are determined using the previously described random sampling approach. This heuristic approach to grasping not only bolsters the stability of grasp actions but also enriches the dataset with a higher proportion of complex interaction modes, such as "grasp to open", enhancing the dataset's diversity and utility for training models to manipulate articulated objects in 'hard' interaction scenarios.

*3. GMM-based Adaptive Sampling* – To foster a wide array of interaction modes within our dataset, we implement GMM-based adaptive sampling inspired by the methodology outlined in [12]. Following the acquisition of $M$ trajectory datasets $\{T_j|j = 1, 2, ..., M\}$ through random and heuristic grasping sampling from previous interactions, we compute the task embeddings $\{z_j|j = 1, 2, ..., M\}$ based on Equation 9. A Gaussian Mixture Model (GMM) prior is constructed from these task

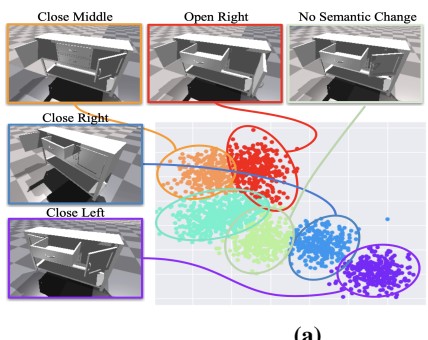
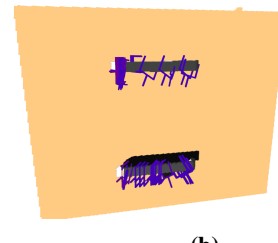

|  (a) | (b) |

**Figure 7: (a) GMM clustering adaptive sampling:** his figure illustrates the visualization of using GMM to represent different interaction modes.
**(b) Visualization of Heuristic Grasping:** We illustrate the proposed grasping using our predefined heuristic with ContactGraspNet [55].

embeddings, denoted as $\mathbb{P}(z|\theta) = \sum_{k=1}^{K} \kappa_k p(a|\theta_k)$, where $\theta_k$ represents the parameters of each Gaussian component within the mixture. The choice of $K$, the number of clusters, is a hyper-parameter that reflects the presumed number of interaction modes inherent to the object.

Subsequently, we cluster the task embeddings $z_j$, assigning a unique cluster label to each corresponding trajectory. We found that the task embeddings from different trajectories grouped within the same cluster indicate a similar interaction mode, as they share proximate visual characteristics from initial and final observation. Upon clustering, a new GMM is formulated for each cluster, based on the action sequences, represented as $\mathbb{P}_k(a|\phi) = \sum_{l=1}^{L} \beta_l p_k(a|\phi_l)$. We then aim to sample an equal number of actions from each cluster, ensuring that the representation of actions—and, by extension, interaction modes—within the dataset are as diverse as possible, thus facilitating a comprehensive exploration of the articulated object's potential interactions.

Utilizing these sampling methodologies, we concurrently collect data across all articulated objects within our dataset, culminating in a dataset denoted as:

$$D = \{T_j\}_{\text{random}} \cup \{T_j\}_{\text{grasp}} \cup \{T_j\}_{\text{GMM}} \tag{10}$$

$$= \{(a_i, O_i)_j\}_{\text{random}} \cup \{(a_i, O_i)_j\}_{\text{grasp}} \cup \{(a_i, O_i)_j\}_{\text{GMM}} \tag{11}$$

$$= \{(O_i, a_i)\}_{\text{random} \cup \text{grasp} \cup \text{GMM}} \tag{12}$$

After data collection, we enrich each trajectory within our dataset by associating the respective task embedding with the data tuple $(O, a)$, thereby forming atomic training data instances represented as $(O, a, \epsilon)_j$.

### 7.4.2 Data Collection Algorithm

**The dataset we developed for training purposes is available on our official website.** Our dataset was constructed through a combination of random sampling, heuristic grasp sampling, and Gaussian Mixture Model (GMM)-based adaptive sampling, featuring the Franka Emika robot engaging with various articulated objects across multiple interaction modes. It encompasses categories such as faucets, tables, storage furniture, doors, refrigerators, and switches, with 8 unique instances per category. For each instance, we collected 150 trajectories, ensuring comprehensive coverage of the objects' interaction modes. Objects were scaled to realistic size and initialized in a 'half-open' state, denoting a median value for each degree of freedom (DoF). The data collection methodology is detailed in Algorithm 1.

### 7.5 Model Architecture and Implementation Details

This section outlines the detailed implementation of the model architecture, encompassing both the mode selector and the action predictor components.

---

**Algorithm 1** Data Collection Algorithm

---

**Require:** Initial observation $O^i$, Number of GMM component $K$, hyper-paramter $M$ for GMM in each cluster
**Ensure:** All sampled trajectories are filtered successful by evaluating $\epsilon > \bar{\epsilon}$

  $D \leftarrow \emptyset$                                       ▷ Set the initial dataset to be empty
  **while** $D$ not have enough data **do**
     $D_r = \{(a, o)_i\} \sim$ RandomSampling                     ▷ Random Sampling
     $G = \{g_i\} \sim$ GraspNet$(O^i)$                    ▷ Sample Grasp using GraspNet
     $D_g = \{(a, o)_i\} \sim$ GenerateTraj$(G)$           ▷ Gnerate trajectory based on grasp
     $D \leftarrow D \cup D_r \cup D_g$
     $\epsilon_i \sim D$                              ▷ Compute task embedding in current $D$
     Cluster $\epsilon_i$ with GMM, assign cluster label on each trajectory
     $\{D_j | j = 1, ..., K\} \leftarrow D$
     $D_{GMM} \leftarrow \emptyset$
     **for** $j$ in range $K$ **do**
        Extract $D_j$ in $D$ based on cluster label
        $p(D_j | \boldsymbol{\pi}, \boldsymbol{\mu}, \boldsymbol{\Sigma}) = \prod_{n=1}^{N} \left( \sum_{m=1}^{M} \pi_m \mathcal{N}(\mathbf{x}_n | \boldsymbol{\mu}_m, \boldsymbol{\Sigma}_m) \right)$      ▷ fit GMM
        $\hat{D}_j \leftarrow \{(a, o)_i\} \sim p(D_j | \boldsymbol{\pi}, \boldsymbol{\mu}, \boldsymbol{\Sigma})$      ▷ Sample action from GMM
        $D_{GMM} \leftarrow D_{GMM} \cup \hat{D}_j$
     $D \leftarrow D \cup D_{GMM}$

---

### 7.5.1 Mode Selector Architecture and Implementation Detail

This section revisits the stochastic variables' definitions and distributions, as previously emphasized. The distributions of the model parameters are formalized as follows:

$$p(c) = \text{Multi}(\pi) \tag{13}$$

$$p(y) = \mathcal{N}(0, \mathbf{I}) \tag{14}$$

$$p_{\xi, \beta}(\epsilon, x, y, c | O^i) = p(y)p(c)p_{\xi}(x|y, c, O^i)p_{\beta}(\epsilon|x, O^i) \tag{15}$$

$$p_{\xi}(x|y, c, O^i) = \prod_{k=1}^{K} \mathcal{N}(\mu_{c_k}(y, O^i), \Sigma_{c_k}(y, O^i)) \tag{16}$$

$$p_{\beta}(\epsilon|x, O^i) = \mathcal{N}(\mu_{\beta}(x, O^i), \Sigma_{\beta}(x, O^i)) \tag{17}$$

Here, $\mu_{c_k}, \Sigma_{c_k}, \mu_{\beta}, \Sigma_{\beta}$ are the model parameters to be optimized. Furthermore, we delineate the generative model and compute the inference at test time by defining the posterior as follows:

$$q(x, y, c | \epsilon, O^i) = \prod_i q_{\psi_x}(x|\epsilon, O^i)q_{\psi_y}(y|\epsilon, O^i)q_{\psi_c}(c|x, y, O^i) \tag{18}$$

This necessitates the computation of three additional network parameters: $q_{\psi_x}, q_{\psi_y}, q_{\psi_c}$. We then elaborate on deriving the posterior $q_{\psi_c}(c|x, y, O^i)$ for categorical variables $c$, employing the Gumbel Softmax for the representation of categorical distributions.

Notice that $c$ is a categorical parameter that $c \sim \text{Multi}(\pi)$. We defined that $c \in \mathcal{C} = \{c_1, c_2, ..., c_k\}$ and the each class probability is described as $\{\pi_1, \pi_2, ..., \pi_k\}$. We use the Gumbel Softax trick which provides a simple and efficient way to draw samples $c$ from a categorical distribution with class probabilities $\{\pi_1, \pi_2, ..., \pi_k\}$. The following form represents the categorical $c$ as,

$$c = \text{one-hot}(\text{argmax}_i[g_i + \log \pi_i]) \tag{19}$$

where $\{g_1, g_2, ..., g_k\}$ are i.i.d samples drawn from Gumbel(0,1). Assuming that categorical samples $c$ are encoded as $k$-dimensional one-hot vectors $\omega$ lying on the corners of the $(k-1)$-dimensional simplex $\Delta^{k-1}$ We use the softmax function as a continuous, differentiable approximation to arg max, and generate $k$-dimensional sample vectors $\omega \in \Delta^{k-1}$. We defined $\omega$ as

$$\omega_i = \frac{\exp((\log(\pi_i) + g_i)/\tau)}{\sum_{i=1}^{k} \exp((\log(\pi_i) + g_i)/\tau)} \tag{20}$$

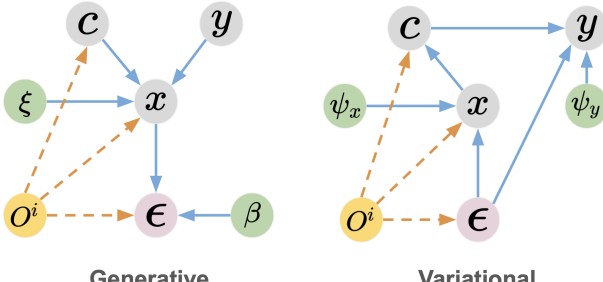

**Generative**            **Variational**

**Figure 8:** The graphical representations elucidate the Conditional Gaussian Mixture Variational Autoencoder (CGMVAE) framework, showcasing two distinct models: the generative model on the left and the variational family on the right. These graphical models serve to visually communicate the structural and functional relationships between variables within the CGMVAE, illustrating the data generation process and the approximation strategy employed by the variational family to infer latent variable distributions.

Where $\tau$ is the temperature as the hyperparameter. Therefore, we define the density of the Gumbel-Softmax distribution as,

$$p(c) = p_{\pi,\tau}(\omega_1, ..., \omega_k) = \Gamma(k)\tau^{k-1} \left( \sum_{i=1}^{k} \frac{\pi_i}{\omega_i^{\tau}} \right)^{-k} \prod_{i=1}^{k} \frac{\pi_i}{\omega_i^{\tau}} \tag{21}$$

Now, given the representation of the categorical distribution of $c$ from Equation 21, we derive how we compute the posterior $q_{\psi_c}$ for $c$. We consider the posterior $q_{\psi_c}(c = c_j | x, y, O^i)$ given $c = c_j$,

$$q_{\psi_c}(c = c_j | x, y, O^i) = \frac{p(c = c_j)p(x | c = c_j, y, O^i)}{\sum_{l=1}^{k} p(c = c_l)p(x | c = c_l, y, O^i)} \tag{22}$$

$$= \frac{\pi_j p(x | c = c_j, y, O^i)}{\sum_{l=1}^{k} \pi_l p(x | c = c_l, y, O^i)} \tag{23}$$

Therefore, we derive the posterior $q_{\psi_c}$ directly and leave 2 posterior network $q_{\psi_x}, q_{\psi_y}$ to be trained.

Based on the following discussion, we draw the generative model and variational model view as graphical models in the Figure 8.

In the implementation detail, we write parameters $p_\beta = (\mu_\beta, \Sigma_\beta)$ and $p_\xi = (\mu_{c_k}, \Sigma_{c_k})$ to generate a Gaussian distribution with each representing the mean and variance. We implement the network $\mu_{c_k}, \Sigma_{c_k}, \psi_x, \psi_y$ with a multi-layer ResNet and implement the network $\mu_\beta, \Sigma_\beta$ as a multi-view transformer since both $O^i$ and $\epsilon$ represent multi-view information with the same number on the channel as the correspondent view number. We show our model $\mu_\beta, \Sigma_\beta$ architecture in Figure 9.

### 7.5.2 Mode Selector Training and Inference

We illustrate the functionality and application of our mode selector through two distinct plots, highlighting both the training process and the inference mechanism for task embedding generation.

Figure Figure 11a depicts the model's operation during training, where it processes the conditional variable $O^i$ along with the ground truth data $\epsilon$, to accurately reconstruct the task embedding.

Conversely, Figure Figure 11b demonstrates the inference stage, where the model, requiring only the initial observation $O^i$ and a discretely sampled cluster (employing an 8-cluster configuration for implementation), successfully generates the corresponding task embedding $\epsilon$.

### 7.5.3 Action Predictor

We provide the architecture of the action predictor which is a joint transformer that takes in task embedding $\epsilon$ and novel view as input. The detailed implementation is shown at Figure 10.

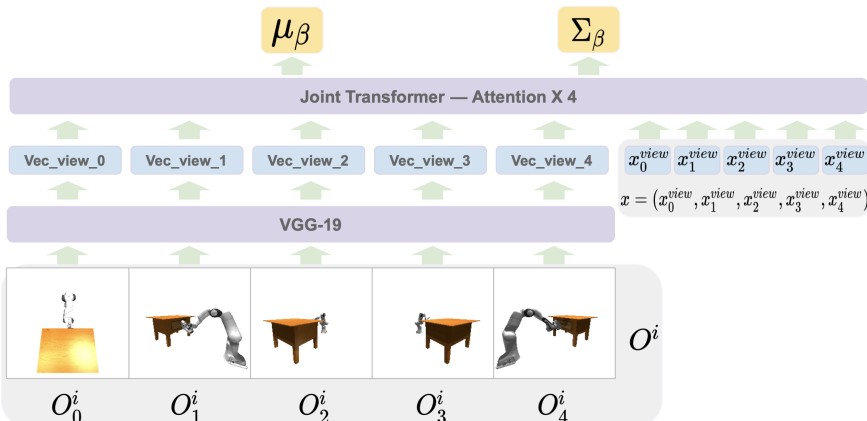

**Figure 9: Mode Selector Decoder Architecture**: The depicted architecture highlights the functionality of the mode selector decoder, which is designed to process two primary inputs: multi-view RGBD images $O^i = (O_0^i, O_1^i, O_2^i, O_3^i, O_4^i)$, and the Mixture of Gaussian (GMM) variable $x$. It is important to note that $x$ can be represented as a multi-view feature vector, with our encoding approach preserving the separation of multi-view channels. Initially, the multi-view RGBD images are passed through a pre-trained VGG-19 image encoder to extract feature vectors for each view. Subsequently, these feature vectors, along with the GMM variable $x$, are inputted into a joint transformer. This transformer, featuring four attention layers, is tasked with producing the means and variances associated with the reconstructed task embedding $\epsilon$.

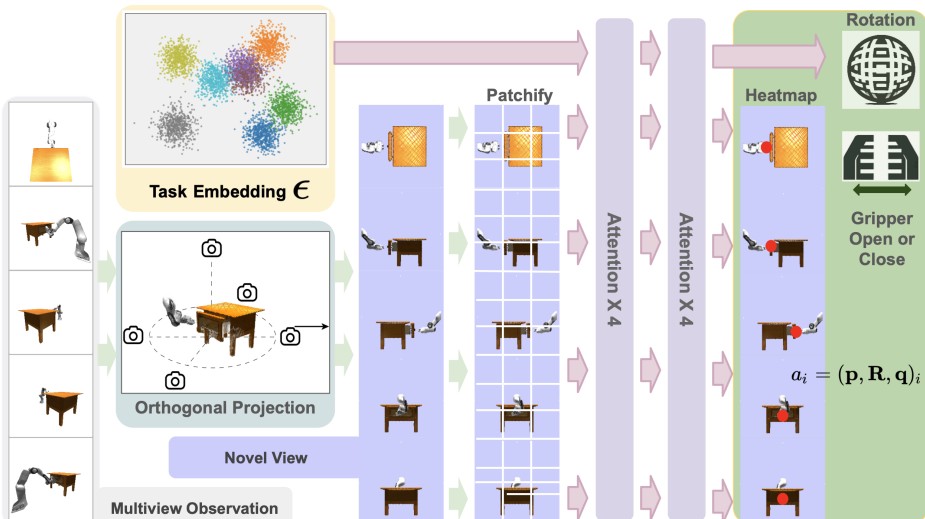

**Figure 10: Action Predictor Architecture**: This model integrates multi-view observations directly as input, sourced from predefined cameras within the scene. The process begins with the extraction of five RGBD images, which are subsequently transformed into RGB point clouds. These are then subject to orthogonal projection to generate five novel view images. Subsequently, these novel views are partitioned into smaller patches and fed into a joint transformer. This transformer, characterized by four attention layers, integrates the sampled task embedding derived from a Mixture of Gaussian distribution. The architecture of the joint transformer encompasses eight attention layers, culminating in the production of a heatmap. This heatmap delineates the action's translation, the discretized rotation, and a binary variable indicating the gripper's state—open or closed.

## 7.6 More Qualitative Results

We supplement our presentation with additional qualitative results, further elucidating the model's proficiency in learning the disentanglement of interaction modes. Initially, we demonstrate the efficacy of the mode selector through a t-SNE plot. This choice of visualization is motivated by our methodology of training the mode selector and action predictor independently, allowing for a focused examination of the mode selector's performance.

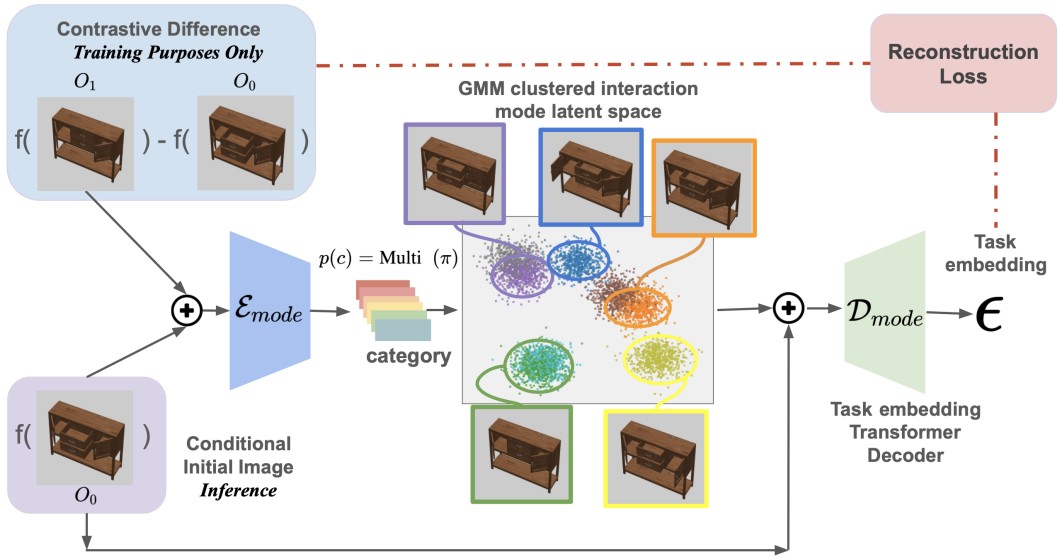

**(a) Training Process of the Mode Selector**: This figure illustrates the training procedure of the mode selector, mirroring the approach of a conditional generative model. It highlights the contrastive analysis between the initial and final observations—the latter serving as the ground truth for task embedding—to delineate generated data against the backdrop of encoded initial images as the conditional variable. The process involves inputting both the generated task embedding data and the conditional variable into a 4-layer Residual network-based mode encoder, which then predicts the categorical variable $c$. Following the Gaussian Mixture Variational Autoencoder (GMVAE) methodology, the Gaussian Mixture Model (GMM) variable $x$ is computed and introduced alongside the conditional variable to the task embedding transformer decoder. This model is tasked with predicting the reconstructed task embedding, sampled from the Gaussian distribution as outlined in the architecture of the mode selector decoder, and calculating the reconstruction loss against the input ground truth data.

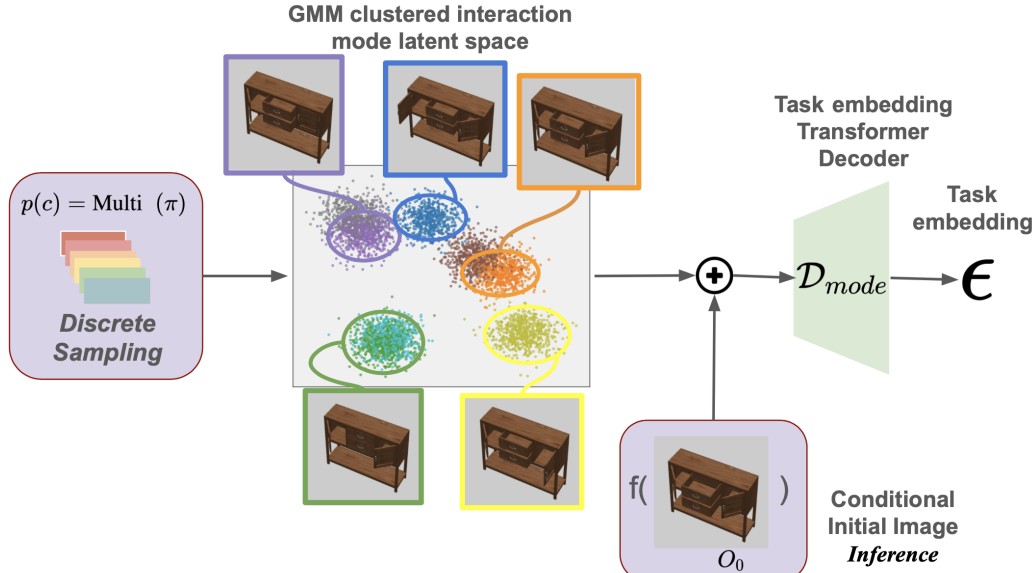

**(b) Inference Process**: In the inference phase, the agent discretely samples a cluster from the trained Gaussian Mixture Variational Autoencoder (GMVAE) model to calculate the Mixture of Gaussian variable $x$. This variable $x$, in conjunction with the conditional variable (initial image observation), is then inputted into the mode selector transformer decoder. The objective is to reconstruct the task embedding for inference, effectively translating the conditional information and sampled cluster into actionable embeddings.

Subsequently, we extend our qualitative analysis with figures akin to those presented in the main paper, offering a comprehensive view of the model's capabilities. These additional figures serve to reinforce the insights gained from the initial results, showcasing the model's nuanced understanding of interaction modes through the distinct visual representations of the data.

### 7.6.1 Mode selector TSNE plot Figure 16

Utilizing our pre-trained Conditional Gaussian Mixture Variational Autoencoder (CGMVAE) mode selector, we conduct disentanglement learning visualization on our comprehensive dataset. Specifically, we focus on the "single drawer" object (object ID: 20411), employing the mode selector to delineate the generated clusters and compare them with the ground truth task embeddings. The data for this visualization is derived from our dataset, and we calculate the task embedding $\epsilon_j$ for each data point as the difference between the initial and final object states, represented by

$$\epsilon_j = v_j^{init} - v_j^{final} = \mathcal{E}_O(O_j^{init}) - \mathcal{E}_O(O_j^{final})$$

.

Subsequently, we employ a t-SNE plot to simultaneously visualize the ground truth and generated task embeddings. In this visualization, distinct colors within the ground truth plot indicate data points originating from different interaction modes. Similarly, varied colors in the generated plot correspond to data points arising from disparate clusters within the Mixture of Gaussians model. Through this approach, we demonstrate that:

1. The ground truth task embeddings $\epsilon$ are distinctly clustered based on the interaction modes.
2. The CGMVAE model effectively generates clusters that categorize data points by their respective categories $c$.
3. The reconstructed data closely aligns with the ground truth data points, with the majority of the clustered data encompassed within the respective ground truth clusters.

This visualization underscores the efficacy of our generative model mode selector in extracting task embeddings for further application in the action predictor, highlighting the model's capability to discern and categorize interaction modes accurately.

### 7.6.2 Action Predictor Qualitative Results

We present extensive qualitative results in Figure 17a, Figure 17b, Figure 18a, and Figure 18b, demonstrating the model's ability to predict distinct interaction modes through discrete sampling. For each object, we explore three different clusters, each representing a unique interaction mode. The initial state of the robot and the articulated object is depicted from three perspectives: top-down, front, and side views. The heatmaps, derived from the top view during manipulation steps, highlight the variance in action space corresponding to different sampled interaction modes. Subsequent imagery illustrates the robot's movement within the simulator and the outcome following interaction with the articulated objects. It is important to note that comprehensive **video demonstrations** accompany this document and are accessible on our website, https://actaim2.github.io/.

### 7.6.3 Comparison of ActAIM2 and VQVAE-RVT

Inspired by the Genie [48] approach, we have compared our ActAIM2 with VQVAE-RVT to assess the efficacy of these models in discerning discrete interaction modes in robotic manipulation tasks. Our primary objective was to evaluate the distinction between interaction modes using a simplified scenario, a single-drawer table, which naturally exhibits two distinct interaction modes: opening and closing.

In our experiments, we visualized the latent spaces generated by both ActAIM2 and VQVAe-RVT. Particularly for VQVAE-RVT, the latent space visualization involved examining the distribution of eight code vectors. As depicted in Figure 12, these vectors clustered into two categories, which ideally should correspond to the two expected interaction modes of the drawer. This clustering pattern was anticipated and desired as it suggests a clear demarcation between the distinct modes of interaction.

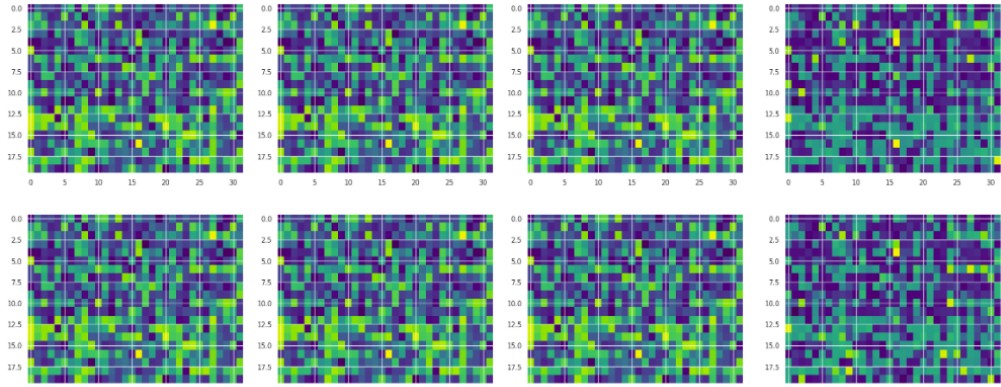

**Figure 12: Visualization of Latent Space Clustering in VQVAE-RVT:** This figure illustrates the distribution of eight code vectors within the latent space, categorized into two distinct clusters. These clusters are intended to represent the discrete interaction modes of opening and closing a drawer. The spatial arrangement highlights the expected separation of code vectors, symbolizing the potential for mode-specific action mapping in robotic manipulation tasks. Despite this apparent clustering, subsequent heatmaps (see Figure 13) reveal a lack of diversity in the action predictions, undermining the practical utility of this model configuration.

However, subsequent visualizations raised concerns about the practical efficacy of the VQVAE-RVT model in our application context. When we explored the heatmaps generated by the VQVAE-RVT model, we observed a critical limitation: all 8 code vectors produced essentially the same heatmap, despite their differing positions in the latent space. This heatmap, illustrated in Figure [Y], consistently depicted all plausible interaction modes for the drawer, regardless of the specific code vector used. This outcome was in stark contrast to the results from ActAIM2, where distinct heatmaps clearly indicated specific interaction actions like pushing or pulling, depending on the sampled cluster within the latent space.

These findings led us to conclude that merely replacing the GMVAE component with a VQVAE in the setup did not achieve the desired disentanglement of interaction modes. The VQVAE-RVT model failed to map the code vectors to unique, mode-specific interaction strategies, instead converging on a generalized representation that was not useful for distinguishing between the actionable options of opening and closing the drawer. Consequently, ActAIM2's ability to discriminate between distinct interaction modes via cluster-specific sampling proves superior in contexts demanding discrete and distinguishable action representations.

We also explain this from mathematics which provide an intuition of why GMVAE perform much better than VQVAE. Referring to Figure 8 and the ELBO loss computation in Equation 18 from the appendix, compared to the VQVAE, the most unusual term in our ELBO is the c-prior term. The c-posterior calculates the probability of assigning a data point to a cluster by measuring the distance between x and each cluster position from y. The c-prior helps reduce the KL divergence between the c-posterior and a uniform prior by adjusting both the clusters' positions and the encoded point x. This term aims to merge clusters by increasing their overlap and bringing their means closer together. However, like other KL regularization terms, it can conflict with the reconstruction term and may become too dominant with more training data. This over-regularization issue, common in traditional VAEs, also affects GMVAE clusters, leading to large, degenerate clusters.

### 7.7 Generation of Demonstration Videos

To illustrate the practical applications and effectiveness of ActAIM2, we generated demonstration videos by employing its inference mechanism. The process involves several key steps:

1. **Generative Mode Selection**: Initially, observations are inputted into the generative mode selector of ActAIM2. This component is responsible for reconstructing the task's latent space, which is modeled as a Mixture of Gaussians. This structure enables discrete sampling of clusters, which represent distinct interaction modes that the robotic system can execute.

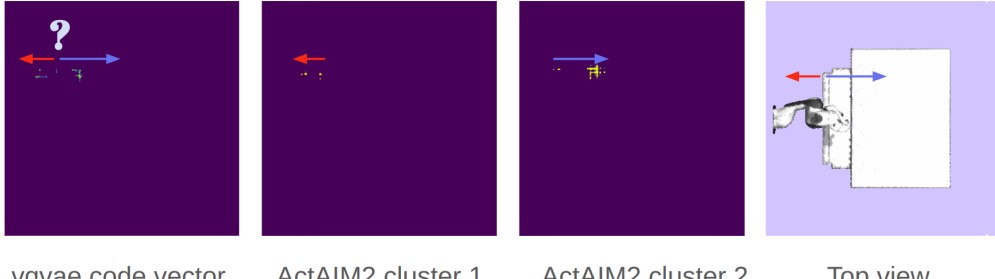

| vqvae code vector | ActAIM2 cluster 1 | ActAIM2 cluster 2 | Top view |

**Figure 13: Comparative Visualization of Action Heatmaps and Observational Data** From left to right: (1) VQVAE-RVT action heatmap synthesized using all eight code vectors, showing identical outcomes across the board, indicating a failure to differentiate interaction modes. (2) Action heatmap generated by \ActAIM2 when sampling from one cluster, demonstrating a specific interaction mode. (3) Action heatmap from \ActAIM2 when sampling from a different cluster, showcasing another distinct mode of interaction. (4) Top-view observation of the drawer, correlating with the spatial contexts of the heatmaps, providing a visual reference for the interaction zones mapped by the heatmaps. This series highlights \ActAIM2's capability to discern and represent distinct action strategies through targeted cluster sampling.

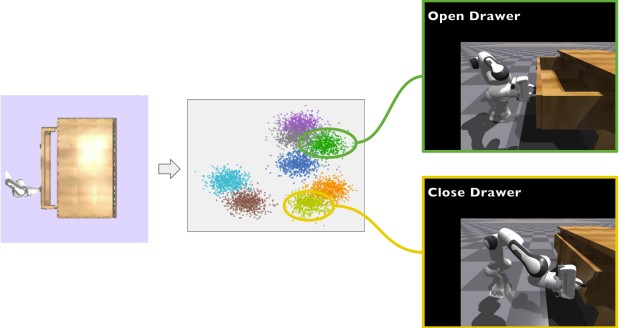

**Figure 14:** Opening and Closing a Drawer: This figure demonstrates the effective action sequence generated by ActAIM2 for a drawer. The left part of the image shows the drawer being opened, showcasing the robot's approach and grip adjustment. The right part of the image captures the drawer in a fully closed position, illustrating the final state after the action sequence execution.

2. **Sampling and Action Prediction**: From the reconstructed latent space, we sample the task embeddings by selecting a cluster within the Gaussian Mixture Model (GMM) and its corresponding Gaussian distribution. This sampled task embedding is then forwarded to the action predictor. The action predictor generates the specific actions needed to interact with the environment effectively.

3. **Simulation and Recording**: As depicted in Figure 14 and Figure 15, ActAIM2 reconstructs an object-based GMM and samples different task embeddings. Depending on the sampled task embedding, different interactions are reconstructed and executed within a simulator. We recorded the manipulation processes, which are detailed in the video provided in the supplementary files. Each video showcases how ActAIM2 navigates through different interaction scenarios, reflecting the diverse capabilities of the model in real-time applications.

This comprehensive demonstration not only validates the functionality of ActAIM2 but also provides a visual understanding of its potential in diverse robotic manipulation tasks. The videos highlight the nuanced interactions achievable through targeted sampling within the model's structured latent space.

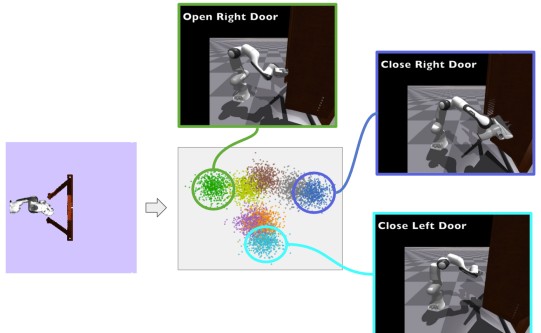

**Figure 15:** Opening and Closing a Door: This figure illustrates the ActAIM2's manipulation capability with a door. The left image displays the door being opened, highlighting the robot's positioning and the initial interaction phase. The right image shows the door completely closed, detailing the end of the manipulation sequence and the effectiveness of the action predictor.

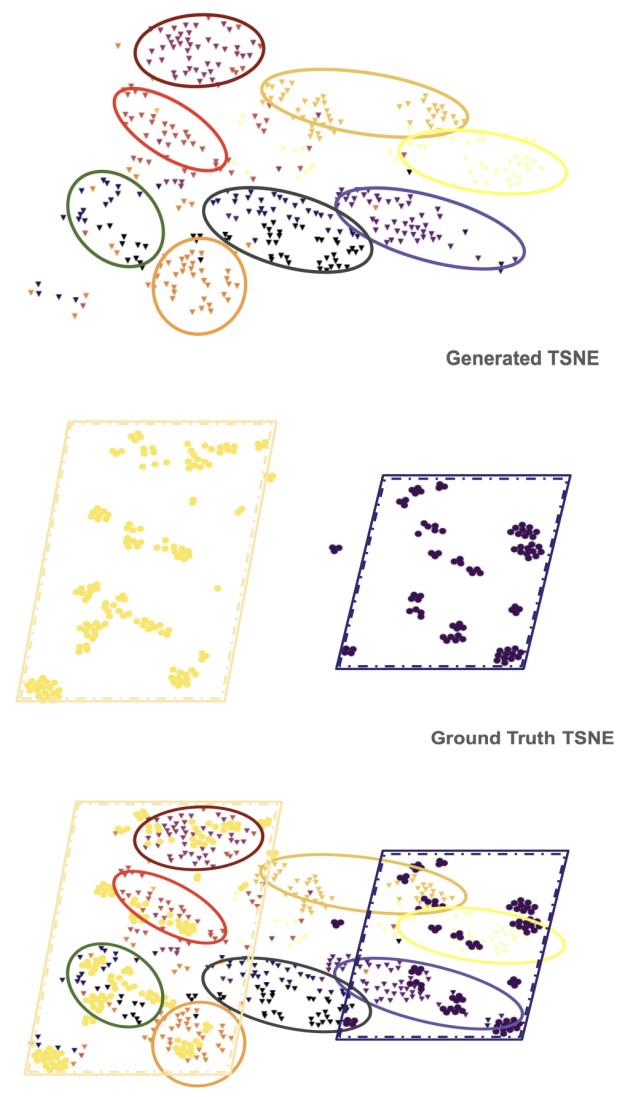

**Generated TSNE**

**Ground Truth TSNE**

**Mixed TSNE**

**Figure 16: Disentanglement Visualization with CGMVAE:** This figure illustrates the efficacy of the Conditional Gaussian Mixture Variational Autoencoder (CGMVAE) in disentangling interaction modes for the "single drawer" object (ID: 20411), using a t-SNE plot for visualization. Task embeddings $\epsilon_j$, defined by the variance between initial and final object states, are visualized in distinct colors to denote various interaction modes and clusters. The sequence of figures demonstrates the CGMVAE's precision in clustering and aligning data points with their respective interaction modes: (1) Generated clusters from the CGMVAE mode selector reveal distinct groupings. (2) Ground truth task embeddings confirm the model's capacity for accurate interaction mode classification. (3) A combined visualization underscores the alignment between generated clusters and ground truth, showcasing the model's ability to consistently categorize tasks within identical interaction modes.

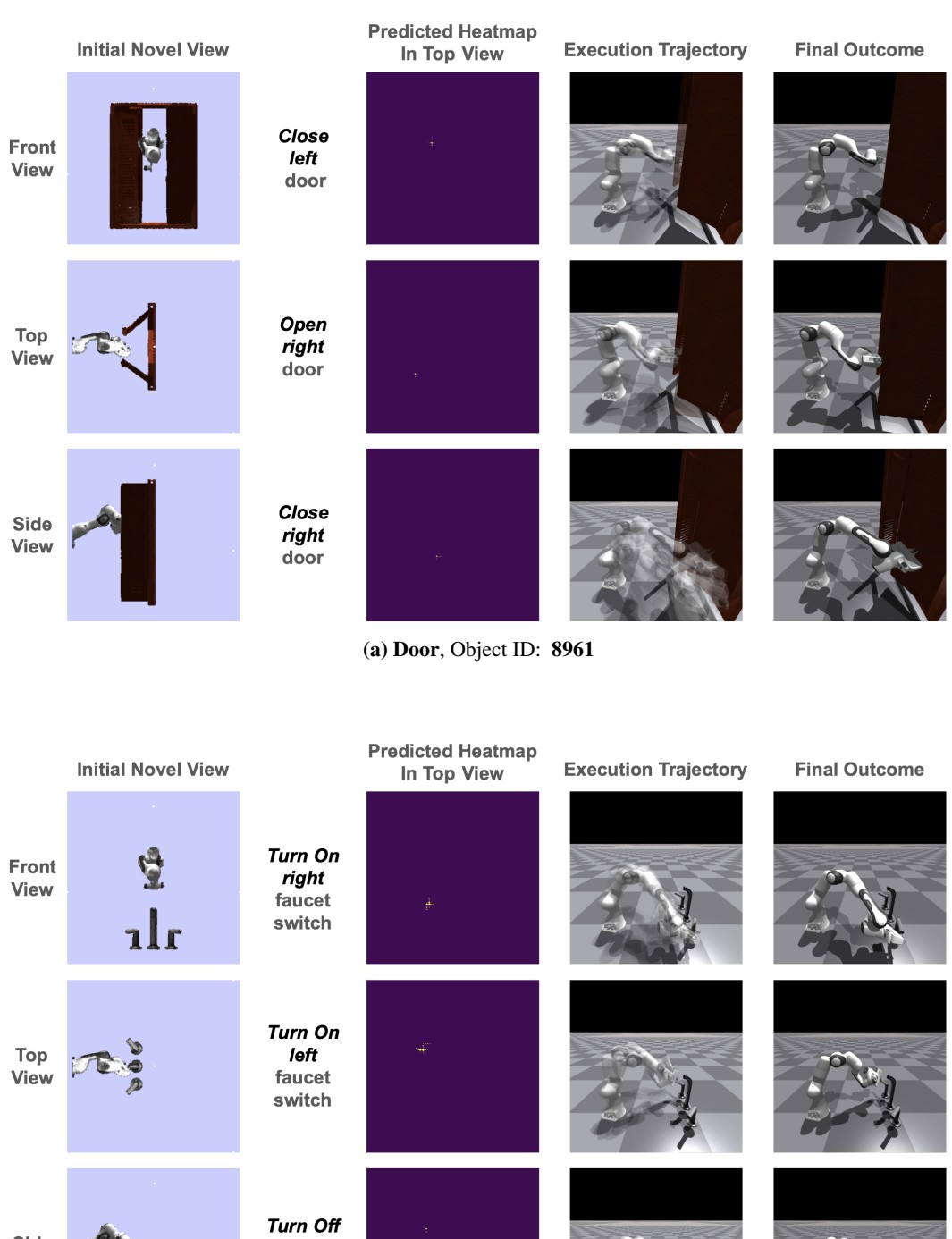

**(a) Door**, Object ID: **8961**

**(b) Faucet**, Object ID: **154**

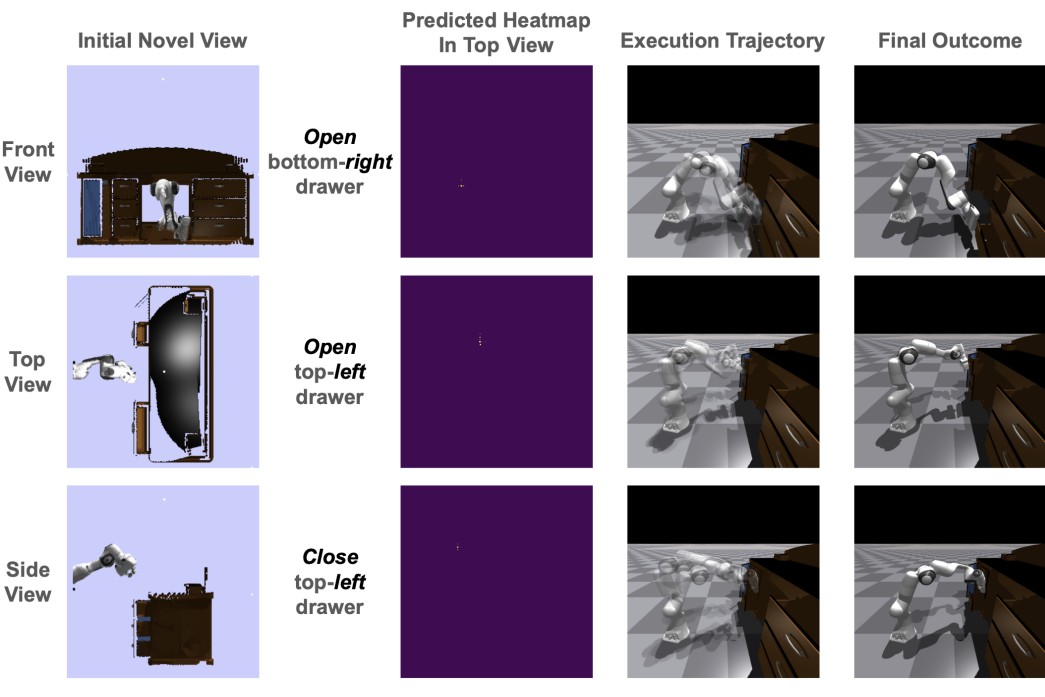

**(a) Table**, Object ID: **19898**

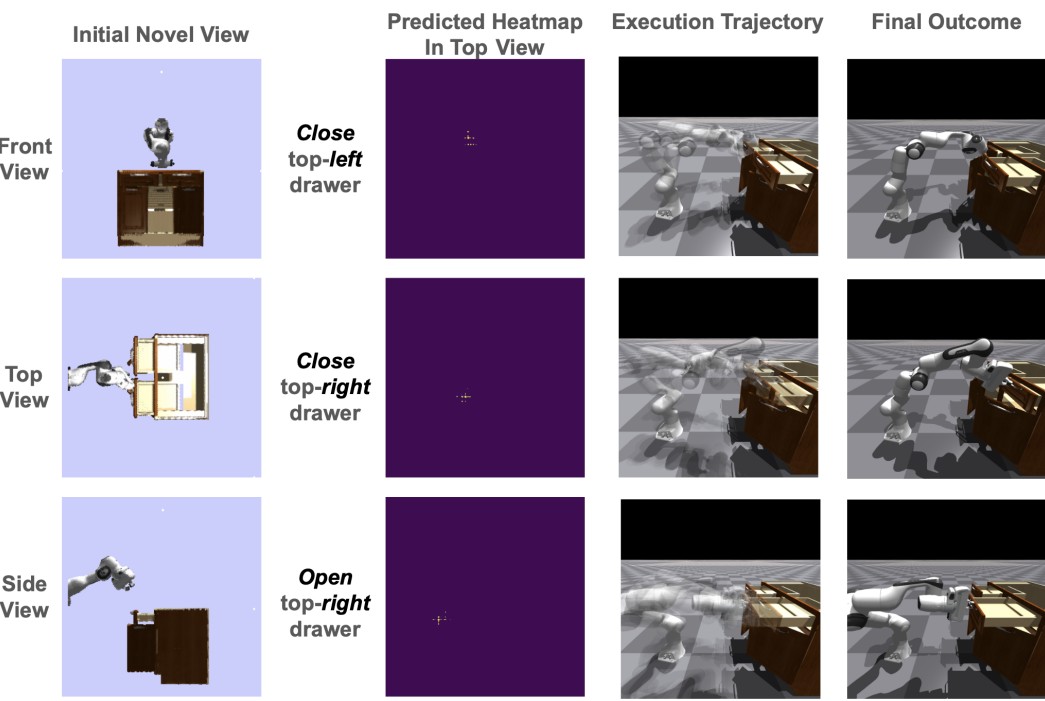

**(b) Storage**, Object ID: **41083**

**Table 3: Object Mode Sampling Evaluation with More Interaction Modes**: We selected objects with more than two interaction modes and evaluated the mode sample success rates using ActAIM2. In practice, ActAIM2 executes actions sampled uniformly across 8 clusters and then reports the corresponding interaction mode success rates. The reported interaction modes are illustrated in Figure 19.

| Object Category | Object ID | 1st Mode% ↑ | 2nd Mode% ↑ | 3rd Mode% ↑ | 4th Mode% ↑ |
|---|---|---|---|---|---|
| ⌐ Faucet | 154 | 65.4 | 16.3 | 6.2 | 0 |
| ⊓ Table | 19898 | 40.1 | 16.2 | 13.2 | 6.3 |
| ▮ Door | 8961 | 79.3 | 43.2 | 14.2 | 10.2 |
| ▦ Storage | 41083 | 55.4 | 26.1 | 17.3 | 0 |

## 7.8 More Experimental Results

In this section, we have added two additional experimental results: object-specific mode sampling evaluation over ActAIM2, where the objects exhibit more than two interaction modes, and average sample success rate evaluation over ActAIM2 with limited perception information. These results aim to demonstrate the stability and robustness of ActAIM2 when predicting outcomes for more complex objects. Furthermore, we demonstrate that ActAIM2 can still accurately predict general outcomes with minimal visual information. This suggests the potential for simplifying the scene setup to adapt ActAIM2 for real-world experiments.

### 7.8.1 Mode Sampling Evaluation on objects with more Interaction Modes

In this section, we assess the mode sampling effectiveness on an object-wise basis as detailed in the experiment section of our paper (see Table 2). We focus on the first and second mode clusters identified by ActAIM2, presenting only two modes because we are examining the average mode sample success rate across categories, typically involving around eight different instances. For simpler articulated objects like a table with a single drawer or a faucet with a single switch, there are typically only two valid interaction modes. This is because these objects are initialized at the mean joint value $j_{init} = (j_{max} + j_{min})/2$. However, most articulated objects feature more than two joints, leading to multiple interaction modes.

To illustrate this, we select more complex objects from various categories, including faucets, tables, doors, and storage units. We first showcase plots depicting the interaction modes for each object from Figure 19. Following this, we present how ActAIM2 predicts the various interaction modes. These objects and their interaction demonstrations are also featured on our website through illustrative videos.

From Table 3, we find that ActAIM2 is capable of predicting multiple interaction modes for more complex objects and achieves a reasonable success rate among the demonstrated interaction modes.

### 7.8.2 ActAIM2 evaluation with less visual information

In the main paper, we utilize five cameras positioned around the articulated object to gather visual information. To assess the model's applicability in real-world scenarios, we evaluate its performance when trained with reduced visual information. Specifically, we conduct two additional experiments: one training with only depth information and another using a single view RGB-D camera. We employ the same dataset as in the experimental section but mask out extra information. For qualitative results, please refer to. Our results demonstrate that ActAIM2 can effectively train with less visual data, with only an acceptable level of performance decrease.

From Table 4, we find that training with only depth information does not significantly affect performance, with less than a 5% decrease in success rate when color is omitted. However, using only a single camera view impacts the success rate more substantially, particularly when the model is applied to larger objects. Scalability also significantly decreases with a single view camera. Nonetheless, performance on previously seen objects remains at a similar level.

**Table 4: Robotic Interaction Mode Discovery with Less Vision** We evaluate ActAIM2 under conditions of reduced visual information to demonstrate its stability and robustness. Specifically, we list experiments where ActAIM2 is trained using only depth data and single-view camera observations. Despite these constraints, it maintains a success rate drop within an acceptable range.

| Test Set | Seen Objects | | | | | | | Unseen instances | | | | | | | Unseen Cats | | | |
|---|---|---|---|---|---|---|---|---|---|---|---|---|---|---|---|---|---|---|
| SSR % ↑ | | | | | | | AVG | | | | | | | AVG | | | | AVG |
| ActAIM2 | 65.3 | 43.2 | 52.1 | 69.2 | 25.3 | 36.2 | 48.6 | 44.9 | 41.2 | 41.5 | 60.2 | 20.1 | 24.4 | 38.7 | 34.3 | 28.9 | 34.1 | 32.4 |
| Depth Only ActAIM2 | 61.5 | 41.9 | 49.3 | 61.5 | 26.3 | 35.1 | 45.9 | 44.3 | 42.6 | 34.2 | 43.7 | 21.4 | 20.4 | 34.4 | 30.2 | 21.2 | 29.5 | 27.0 |
| Single View ActAIM2 | 55.3 | 32.1 | 35.2 | 45.3 | 21.4 | 27.9 | 36.2 | 32.5 | 26.1 | 29.7 | 32.1 | 15.3 | 13.2 | 24.8 | 15.3 | 13.2 | 12.3 | 13.6 |

## 7.9 Comparison between Prior Works

### 7.9.1 Comparison between ActAIM [12]

We state that the main differences between our work and [12] lie in 2 major aspects which are data collection (environmental setup) and method (discrete representation).

Data Collection For ActAIM, we utilized a floating parallel-jaw gripper for data collection, eliminating concerns related to invalid collisions or reachability issues. The action primitives learned by ActAIM include identifying the point to graspor interact and determining the direction to move post-grasp, akin to the approach used in Where2Act [43].

For ActAIM2, we employed a full Franka Emika robot, necessitating consideration of workspace constraints and potential collisions with the robot body. Directly applying the GMM adaptive data collection method from ActAIM proved challenging; hence, we incorporated a heuristic grasping strategy to aid in identifying graspable parts of the object. We developed an analytical technique to determine which segments of the point cloud are likely to correspond to handles on articulated objects. This was followed by using DBSCAN to identify and filter potential heuristic handles. Subsequently, we employed ContactGraspNet [55] to predict possible grasping points, as illustrated in Figure 5 of the Appendix. We combined these heuristic grasping actions with randomly sampled data to execute GMM adaptive sampling, similar to ActAIM, ultimately integrating these components into our final dataset.

Method Both ActAIM and ActAIM2 share the approach of decomposing the policy into a mode selector and an action predictor. The key difference lies in how they represent the interaction mode. In ActAIM, a simple Conditional Variational Autoencoder (CVAE) is used as the mode selector to represent the interaction mode for manipulating articulated objects. However, the latent space in this approach forms a Gaussian distribution, which cannot be easily sampled or clustered.

In Section 5.3, we experimented on interaction mode grounding, demonstrating that ActAIM struggles to associate task embedding with a specific interaction mode. The purpose of this experiment was to show that if the structure of the latent space is well-organized (meaning that all interaction modes are distinctly learned), the grounding should converge after a few interactions. The results revealed that ActAIM fails to provide a clear and discrete representation of interaction modes.

Furthermore, we visualized the interaction mode representation from ActAIM, as shown in Figure 14 of the Appendix. The visualization indicates that the interaction modes learned by ActAIM are noisy and disorganized, with different modes scattered throughout the space. This explains why the experiment in Section 5.3 did not successfully ground specific interaction modes.

### 7.9.2 Comparison between works with affordance prior

Interaction modes are characterized by significant visual changes in the observation space. In contrast to methods such as SayCan [7], VADER [56], or RLAfford [57], our approach focuses more on visual affordance without incorporating action data. The interaction mode seeks to predict and categorize the limited future states of articulated objects, such as opening or closing a partially open drawer. Ideally, the mode selector $p(\epsilon|o)$ could function as the encoder in a video generative model. In the context of articulated object manipulation, where visual information is readily available,

our mode selector model is trained using a pre-trained visual encoder (VGG-19 in our case). This mode selector could be substituted with any world model or video prediction model that forecasts possible future states. Ultimately, the interaction mode infers a distribution over plausible future states achievable through interaction, based on an input image of a scene. For articulated objects, any visual change in the object, especially if it occludes the robot, is significant. We aim to extract and cluster such information to facilitate learning in the action decoder.

## 7.10 Limitations

In this section, we discuss 3 major limitations of the current ActAIM2 which are imbalanced success rate in different interaction modes, lack of general scalability, and limited to articulated object manipulation. Moreover, we provide potential solution to all these limitations.

### 7.10.1 Imbalanced Success Rate

The experimental results demonstrate that while ActAIM2 can represent different interaction modes using discrete clusters, the success rates among these modes are imbalanced. The Mode Sampling Evaluation table shows that the average success rate in the second mode cluster is much lower than in the first mode cluster for ActAIM2. Additionally, Figure 6b illustrates this imbalance using a half-open drawer demonstration: pushing the drawer in has an over 80% success rate, whereas opening the drawer only achieves about a 20% success rate.

This imbalance in success rates could impact the application of ActAIM2, particularly in interaction mode grounding and long-term planning. As shown in section 5.3, the interaction mode grounding experiment using RL required nearly 50 iterations to distinguish between closing and opening modes, with the average success rate for opening still not being high enough.

In more complex scenarios with multiple articulated objects, this imbalance might affect the planning process if ActAIM2 is used as a prior. It can be difficult to determine whether the sampled interaction mode will lead to a "nothing happens" outcome or a rare interaction mode. Therefore, multiple sampling attempts are needed, reducing the efficiency of data use.

Based on our experience with model training, we believe that an imbalanced dataset is the cause of this issue. Although we use the GMM adaptive data collection method to increase interaction mode diversity, the challenge remains that modes requiring precise grasping need more accurate action labels. To address the problem of imbalanced success rates, we propose collecting more stable data. Specifically, we suggest incorporating more expert tele-operation control data into the dataset to achieve more reliable grasping, rather than relying solely on completely self-generated data.

### 7.10.2 Lack of General Scalability

In the experimental table, we observed a significant decrease in success rates from seen objects to unseen objects, and further to unseen categories. The success rate for generalizing across different categories dropped by nearly 50%. While ActAIM2 shows some level of scalability, it performs poorly when scaling across different categories.

To address this scalability issue, we suggest training our model on more data. Currently, ActAIM2 is trained on only 6 categories with 8 unique instances, and we have collected 150 trajectories for each articulated object. We believe that incorporating a greater variety of object categories is necessary to better demonstrate scalability across different types of articulated objects.

### 7.10.3 Limited to Articulated Object Manipulation

ActAIM2 is designed by decomposing the policy into a mode selector and an action predictor, where the mode selector defines distinct interaction modes. In this work, we define an interaction mode as a significant visual change in the articulated object, assuming that any visual change is meaningful if it pertains to the object. However, this definition doesn't hold true for tool manipulation tasks. For

example, in a hammering task, simply moving the hammer around on the table without grasping it is meaningless, yet it would still qualify as an interaction mode under the current definition.

This misalignment leads to inefficiencies during training when applying ActAIM2 to tasks like hammering. Additionally, data collection becomes a significant challenge due to the longer task duration and the need for more precise actions.

To address these issues, we suggest first improving the data collection process and redefining the concept of interaction modes for tool manipulation tasks. Specifically, we should introduce stable grasping as an additional interaction mode and utilize large video-language models to assess whether the resulting visual changes are meaningful.

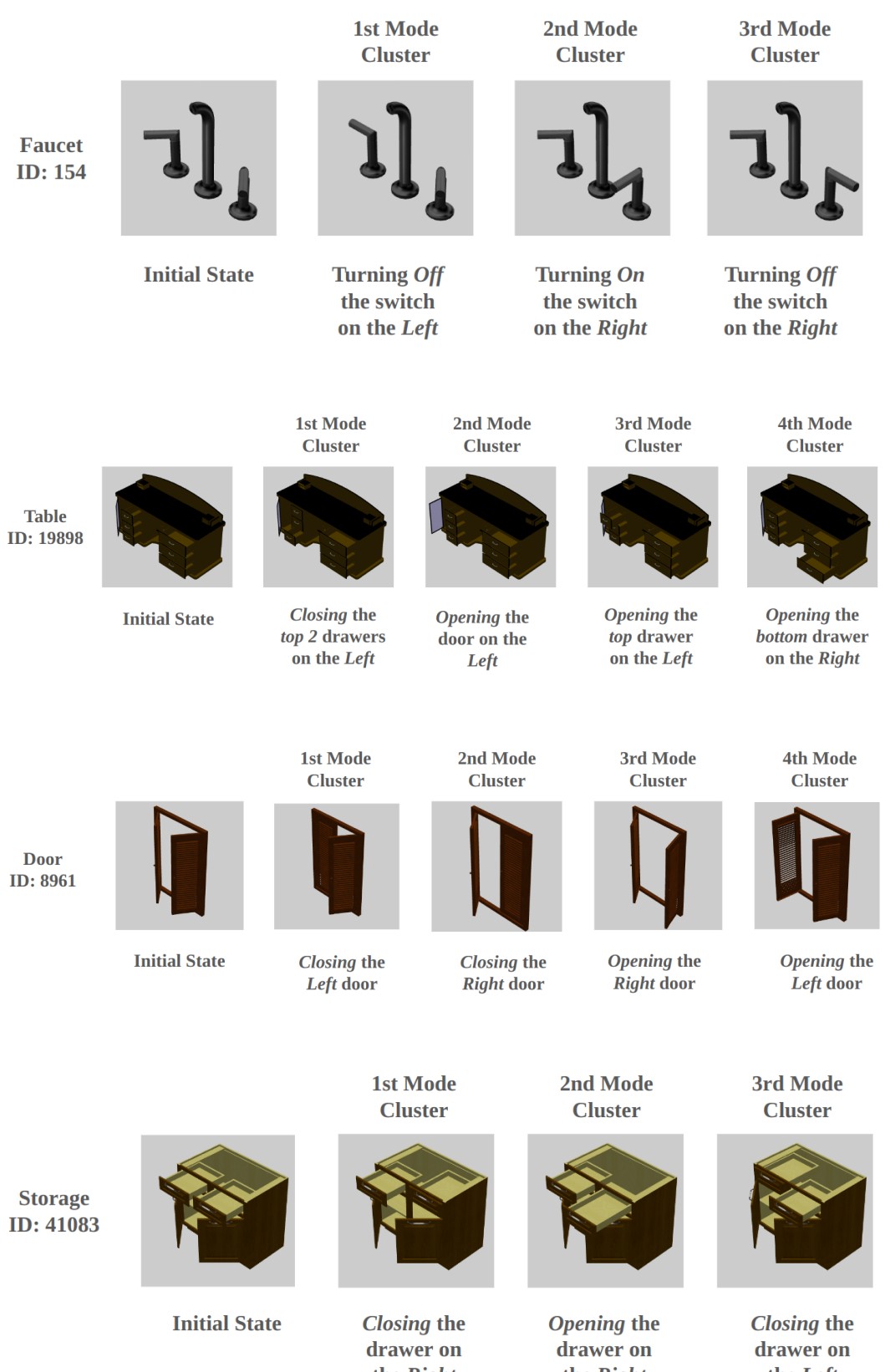

**Figure 19:** We illustrate objects from various categories that exhibit more complex interaction modes. These objects are qualitatively evaluated based on their sample success rates for each presented interaction mode. Video demonstrations of manipulating these objects are also available on our website.

