# OpenReview forum: "Discovering Robotic Interaction Modes with Discrete Representation Learning"
_robot-learning.org/CoRL/2024/Conference — CoRL 2024_

### Official Review · Reviewer_DsEP · 2024-07-20
**Review for Submission461**

**Originality:** 2
**Technical Quality:** 3
**Clarity Of Presentation:** 3
**Potential Impact:** 2
**Recommendation:** 3
**Confidence:** 4

**Review:**

- **Clarity** - The paper presents an interesting idea but the method is difficult to follow. . The paper would benefit significantly from a clearer, more simplified explanation of the method and the experiment design.
- Originality and significance - Fair
- **Strengths**
    - I like the concept of combining object interactions with robot manipulation which has been used as a prior to enable autonomous data collection! This shows potential to improve the efficiency for robot manipulation.

- **Weaknesses**
    - Real world applicability may be limited - I am skeptical of how this method would fare on a real robot. While the concept of self supervised data generation already exists - my understanding of the data collection method is that it allows the robot to fail and then filters out the failures heuristically.
    - The priors used are handcrafted, please clarify the usage of self supervised method - is it the autonomous data collection?
    - While the method is interesting, the paper is hard to follow and could improve with better writing.

**Quality Of The Limitations Section:**

1

**Questions For Rebuttal:**

- Would the authors have any intuition to why the VQ-VAE learnt representations much worse than the GMM vae repesentations? Could this be attributed to codebook initialization?
- It is unclear if the method is self supervised, when all interaction modes are provided. Making the distinction in the paper would improve clarity.
- How are interaction modes significantly different from using affordances as priors? (Additional citations could help.)
- Figure 2: typos - multiview transformer(figure); mode selector(caption)

**Robotics Focus:**

4

**Summary Of Paper:**

ActAIM2 is a manipulation method learns a discrete representation of robot manipulation interaction modes in an unsupervised manner, without expert labels or simulator-based privileged information. The method consists of an interaction mode selector and a low-level action predictor. The selector generates discrete representations of potential interaction modes, while the predictor outputs corresponding action trajectories.

**Summary Of Recommendation:**

I like the method presented but it is explained poorly. The current limitations section could also be improved.

---

### Official Review · Reviewer_ituT · 2024-07-22
**Needs improvement**

**Originality:** 2
**Technical Quality:** 4
**Clarity Of Presentation:** 3
**Potential Impact:** 2
**Recommendation:** 3
**Confidence:** 4

**Review:**

The paper is very well written. The motivation, method, and data collection are well presented, and the experiments include many objects and comparisons to support the paper. The limitations due to the work’s assumptions are clearly explained. Furthermore, it is positive that the data is shared with the community. I suggest the authors also open-source the models for reproducibility.

Major concerns:

The novelty seems limited for this venue compared to ActAim [12]. After checking [12], I believe the major changes are the model updates. Instead of training a CVAE, a CGMVAE is trained, and the action predictor is updated with a transformer-based model that can handle multiple views. The main paper should clearly state the contribution on top of [12].

The method is not evaluated in the real world, which may raise concerns about its applicability. The reasoning behind the lack of real robot experiments should also  be stated.

Other improvements:

Please explain the failure cases during the experiments, whether grasp, execution, or mode selection fails.

The limitations of the models should be stated as well. How would different colors, different instances, and different sizes of objects affect the behavior? Currently, it seems that it can somewhat generalize but a more detailed explanation is needed.

The discussion section only summarizes the results. Explain the relation between design choices of approaches and performance differences. It does not need to be as detailed as the VQVAE-RVT comparison in the Appendix, but at the moment, it is not clear why ActAim2 is better than the baselines.

Please explain unintroduced symbols on page 4 and fix typos ‘/algoname’ in Fig. 11.

**Quality Of The Limitations Section:**

2

**Questions For Rebuttal:**

What are the differences between this work and [12] in the method and the data collection?

How would color change in the environment affect the model? Can different visuals of objects be in extrapolation regions of the models?

Is the pre-trained encoder used in CGMVAE?

What is a different instance of the object? What changes?

What are the second modes for one joint object, such as a faucet, drawer, and door?

What is the percentage of failures during random sampling in data collection?

**Robotics Focus:**

3

**Summary Of Paper:**

In this paper, the authors proposed a method to discover action modes for articulated objects and manipulate these objects using these modes. The authors trained a transformer-based Conditional Gaussian Mixture Variational Autoencoder (CGMVAE) to learn the action modes after collecting data by defining actions as moving the end effector between 4 poses: initiation, reaching, grasping, and manipulating. During data collection, grasping is handled by a pre-trained model, and a Gaussian Mixture Model(GMM)-based adaptive sampling is used to ensure the diversity of the data to discover modes. The success is defined by enough visual change between initial and end images. Afterward, a transformer-based action selector predicts actions using the interaction modes. The authors conducted simulator experiments to verify the model’s capability in comparison with recent methods [3, 43, 12] in the literature. The video only includes robot executions in simulation.

**Summary Of Recommendation:**

My questions are mostly answered. I updated my score.

---

### Official Review · Reviewer_BZdG · 2024-07-22
**Interesting premise, good execution, although could be presented better**

**Originality:** 3
**Technical Quality:** 3
**Clarity Of Presentation:** 2
**Potential Impact:** 2
**Recommendation:** 3
**Confidence:** 3

**Review:**

## Summary

In this work, the authors present ActAIM-2, which is a clustering based method to understand different manipulation modes in objects that can be manipulated in multiple different ways, and then using the discovered manipulation modes to complete the (conditioned) manipulation task. The algorithm works in two stages: the first stage is to train the manipulation mode conditioning, which is trained via a clustering of difference between the first and last frame embeddings. Once these modes are discovered and clustered, the next step is to train a manipulation-mode conditional action predictor model. This model predicts actions in the next-best-pose style, thus bypassing the sequential-ness of most manipulation problem formulations.

The authors then evaluate this algorithm on manipulable objects on the SAPIEN dataset. Compared to the baseline, their algorithm shows higher success rate in both pure success rate and also on mode-specific success rate. Moreover, they show success in sampling modes and grounding the modes using reinforcement learning.

## Strengths

1. I thank the authors for listing out the assumptions, which is a good step to motivate future work.
2. Using GMM is simple, and it seems to work; not everything needs the most complicated algorithms.
3. Interaction mode discovery over multiple object types seem to work well, which shows the robustness of this algorithm.

## Weaknesses
While the paper reads interestingly, there are some anomalies in the result reporting that makes it less appealing. One is not reporting the VQ-VAE-RVT results in Table 1 (line 250-251)

> We did not report the VQVAE-RVT results here since VQVAE-RVT outperforms around 5% in each test compared to ActAIM2 averagely. However, despite the average sample success rate, we show that VQVAE-RVT does not meet our requirement for252
discrete sampling in the following two experiments.

The authors do not clearly mention the discrete mode sampling/RL fine-tuning as a goal too much in the beginning, and it feels like a post-facto justification to disqualify VQVAE-RVT and champion their own method. As a scientist I would like to know how much work was put in improving VQ-VAE RVT sampling, because the VQ-VAE codebooks are generally not of the same structure as normal scalar space, and so special care may be needed for proper sampling.

Secondly, relying on only SAPIEN simulator is slightly unsatisfying, since the physics there is not the most grounded in physics. It would be better to have some real world experiments as well.

Thirdly, the algorithm assumes multi-view RGB-D access which seems difficult to set up in the real world.

Finally, naming the method ActAIM2 seems somewhat anonymity breaking, given that ActAIM exists already.

**Quality Of The Limitations Section:**

2

**Questions For Rebuttal:**

1. What are the blockers to applying this in a real world settings? Could the sampling based data collection be replaced by human demo collection in the real world?
2. What are the authors' opinion on why VQ-VAE sampling fails to get a second mode?
3. Why stop sampling modes at two? Shouldn't there be objects with more than two modes of interaction?

**Robotics Focus:**

3

**Summary Of Paper:**

Clustering-based approach to manipulating articulated objects with multiple articulation mode

**Summary Of Recommendation:**

I do not feel strongly about this work, and I wonder how it can be translated into reality. I am willing to change my mind based on a solid recommendation for real-world deployment.

---

### Author Rebuttal · Authors · 2024-08-12

We appreciate the feedback from the reviewers. Reviewer BZdG highlighted our method as interesting, simple, and elegant, while Reviewer ituT praised our clear writing and presentation of experiments. Reviewer DsEP noted our paper's potential to enhance data efficiency in robotics.


To address feedback on our model, we've **enhanced the paper with an appendix detailing real robot experiments, additional results, comparative analyses, and limitations. A video showcasing ActAIM2's real-world tasks is also included.**
Reviewers noted our strong simulation performance and requested real-world validation, which we've met by incorporating experiments showing ActAIM2's 75% success rate in real-world drawer manipulation, initially developed on simulated data and fine-tuned with real-world data. Further details and demonstrations are available in the appendix and on our website.


Addressing reviewer inquiries on the distinction between ActAIM2 and ActAIM, despite similar policy components like the mode selector and action decoder, ActAIM2 marks a significant evolution with its title "Interaction Mode with Discrete Representation." ActAIM2 shifts from ActAIM's Conditional Variational Autoencoder—which produced a complex Gaussian-distributed latent space—to a discrete representation that simplifies sampling and clustering, enhancing mode distinction and execution. This advancement notably progresses the field of visual affordance by focusing on interaction modes defined by visual changes instead of action data, setting it apart from models like Saycan, Vader, and RLAfford. ActAIM2's mode selector not only predicts future states but also tokenizes actions based on visual inputs, similar to Genie [4] but tailored for action space. This approach enhances interaction with complex environments and is further supported by our GMVAE structure, which, unlike VQVAE, employs dynamic clustering with a c-prior to better handle complex data distributions, reducing KL divergence more effectively which is further discussed in the GMVAE paper.


Overall, ActAIM2 represents a major advancement in robotic learning, introducing a self-supervised approach for precise, discrete sampling of interaction modes, setting it apart from previous models. Supported by innovative data collection and a unique dataset for the Franka robot, comprehensive testing shows ActAIM2's superior performance in both simulations and real-world applications, significantly enhancing data efficiency in robotics.

---

### Decision · Program_Chairs · 2024-09-04

**Decision:**

Accept

**Comment:**

This work introduces a variant of ActAIM that uses a CGMVAE to represent action modes and that can handle multiple views. It appears to be conceptually simple, quite robust and effective in simulation. However, the reviewers perceive this work as rather incremental. It does not clearly state how exactly ActAIM2 differs from ActAIM, and it is unclear how it would fare in the real world. All reviewers ask questions about evaluation. A systematic ablation study would help. Also, Reviewer BZdG asks an important question about the relation to VQVAE-RVT.

As part of the rebuttal, the authors added an appendix describing real-world experiments, additional results, comparisons and limitations. It also satisfactorily answers many questions of the reviewers. The most important points should be integrated into the final version of the paper, if accepted.